# Stress routes clients to the proteasome via a BAG2 ubiquitin-independent degradation condensate

Daniel C. Carrettiero [1,2], Maria C. Almeida [1,2], Andrew P. Longhini[1], Jennifer N. Rauch[1], Dasol Han[1], Xuemei Zhang[1], Saeed Najafi[3], Jason E. Gestwicki [4] & Kenneth S. Kosik[1✉]

The formation of membraneless organelles can be a proteotoxic stress control mechanism that locally condenses a set of components capable of mediating protein degradation decisions. The breadth of mechanisms by which cells respond to stressors and form specific functional types of membraneless organelles, is incompletely understood. We found that Bcl2-associated athanogene 2 (BAG2) marks a distinct phase-separated membraneless organelle, triggered by several forms of stress, particularly hyper-osmotic stress. Distinct from well-known condensates such as stress granules and processing bodies, BAG2-containing granules lack RNA, lack ubiquitin and promote client degradation in a ubiquitin-independent manner via the 20S proteasome. These organelles protect the viability of cells from stress and can traffic to the client protein, in the case of Tau protein, on the microtubule. Components of these ubiquitin-independent degradation organelles include the chaperone HSP-70 and the 20S proteasome activated by members of the PA28 (PMSE) family. BAG2 condensates did not co-localize with LAMP-1 or p62/SQSTM1. When the proteasome is inhibited, BAG2 condensates and the autophagy markers traffic to an aggresome-like structure.

[1] Neuroscience Research Institute, Department of Molecular, Cellular, and Developmental Biology, University of California, Santa Barbara, CA, USA. [2] Center for Natural and Human Sciences, Federal University of ABC, São Bernardo do Campo, SP, Brazil. [3] Department of Chemistry and Biochemistry, University of California, Santa Barbara, CA, USA. [4] Institute for Neurodegenerative Disease, Department of Pharmaceutical Chemistry, University of California San Francisco, San Francisco, CA, USA. ✉email: kosik@lifesci.ucsb.edu

Stimulus responsive phase separation in cells includes liquid-liquid phase transitions due to sharp changes in the physical properties of component proteins and RNA that establish a boundary between the dilute and dense phases[1]. The dense phase can be a biomolecular condensate in which the components collaborate to implement a cellular function. In the context of cellular stress, condensates can provide a safe harbor while adverse conditions exist. During cellular stress most translation shuts down and mRNAs are sequestered in a manner that allows their restoration when the stress is relieved. Hence, many studies have focused on membraneless organelles such as stress granules, processing bodies, and RNA granules, all of which are characterized by the presence of both mRNAs, ribosomes and RNA-binding proteins. While these organelles maintain RNA in a non-translatable state ready for release with relief from stress, the role of these organelles in protein degradation appears to be limited to ubiquitination which has a myriad of both degradative and non-degradative functions. Less attention has been paid to cellular complexes that directly serve protein degradation in the context of stress, particularly the important category of proteins that undergo ubiquitin-independent delivery to the proteasome and operate quasi-independently of the autophagy system. One effort in this direction was the identification of membraneless organelles that implement proteasomal-mediated degradation in nuclear foci under conditions of acute hyperosmotic stress[2]. More generally, ubiquitin-independent degradation tends to be associated with aged proteins and the conformational rigidity of proteins involved in neurodegeneration.

We report herein that Bcl2-associated athanogene 2 (BAG2) condensation assumes a pivotal role in degradation decisions in the context of hyperosmotic stress. BAG2 family members interact with the ATPase domain of Hsp70-Hsc70 chaperones through their BAG domains[3–5]. Importantly, the BAG domain of BAG2 has a low level of homology with the other BAG domains due to a novel dimeric structure in the Hsp70-Hsc70 binding mode[6] as well as a unique ubiquitin-independent role in proteotoxic stress. As client proteins cycle through rounds of folding, BAG2 can inhibit the carboxy-terminus of Hsc70-interacting protein (CHIP) thus preventing ubiquitination of the client and premature delivery to the proteasome[7,8]. However, refolding may not be possible and in this circumstance, BAG2 promotes degradation via the ubiquitin independent proteasome pathway[9]. Full-length BAG2 forms, in vitro, tetramers and higher-order oligomers observed in high molecular weight complexes[6]. Esposito et al.[10] recently reported BAG2 within cytoplasmic particles that contained about 600 other proteins. BAG2 and Hsc70 recognize hydrophobic patches flanked by positively charged residues on one of the most studied client proteins, the first nucleotide-binding domain of the ion channel (NBD1) in cystic fibrosis transmembrane conductance regulator (CFTR)[7]. Oligomeric complexes in which binding properties involve patches of hydrophobic and charged amino acids are critical features of the multivalent interactions utilized for phase separation of condensates[11–13].

Here we found that phase-separated membraneless organelles characterized by BAG2 condensation represent a proteostatic-proteotoxic organelle distinct from stress granules and processing bodies that is highly responsive to hyperosmotic stress and other stressors. These condensates are neither associated with RNA nor ubiquitin, but are associated with the proteasome and chaperones. BAG2 can condense at sites on the microtubule where it can target Tau to the proteasome in a ubiquitin independent manner. BAG2 condensates did not co-localize with LAMP-1 or p62-SQSTM1. When the proteasome is inhibited, BAG2 condensates can traffic to an aggresome-like structure.

## Results

**BAG2 condensation in response to stress.** In stably expressing clover-BAG2 SH-SY5Y cells, less than 50% of the cells contained large numbers of condensates while the remaining cells contained a diffuse BAG2 appearance (Fig. 1a). For most of the cells, these two states were readily distinguishable with very few cells falling in an intermediate range. The average number of condensates was $218.9 \pm 74.1$ per cell with a mean diameter of $0.86 \pm 0.20$ μm (Fig. 1b). These condensates were motile, underwent fusion (Fig. 1c; Supplementary Movie 1), and dynamically exchanged BAG2 molecules with the dilute phase within a time frame indicative of LLPS (Fig. 1d) as demonstrated by FRAP (Fluorescence Recovery after Photobleaching). Hyperosmotic stress (125 mM, sucrose) induced further BAG2 condensation (Fig. 1e, Supplementary Movie 2). Under these conditions, clover-BAG2 condensation increased $4.5 \pm 3.4$-fold ($p < 3.0 \times 10^{-2}$) in total area after one minute and reached a plateau after 10 min with a $22.6 \pm 0.1$-fold increase in total area at 14 min ($p < 1.0 \times 10^{-4}$) (Fig. 1f). Hyperosmotic stress also increased the percentage of cells containing clover-BAG2 condensates from $42.7 \pm 11.7\%$ to $77.4 \pm 11.4\%$ after 14 min ($p = 4.0 \times 10^{-4}$; Fig. 1g). Dissolution of BAG2 condensates occurred in less than two minutes following recovery from hyperosmotic stress (Supplementary Fig. 1a). Other stressors also induced BAG2 condensates over different time intervals: MG132 (30 min), oligomycin (80 min), temperature (42 °C, 30 min) and lipopolysaccharide (LPS; 20 min) (Supplementary Fig. 1b–e).

Staining of endogenous BAG2 in SH-SY5Y cells revealed numerous small puncta distributed throughout the cytoplasm (Fig. 1h, left image). These BAG2 puncta appear to be phase separated, but experimental validation of LLPS of endogenous protein was not technically feasible. After hyperosmotic stress (sucrose, 30 min, 125 mM, Fig. 1h), the average number of endogenous BAG2 puncta per cell did not change significantly (Fig. 1i), but their diameters increased from $0.28 \pm 0.13$ μm to $0.36 \pm 0.19$ μm ($p < 1.0 \times 10^{-4}$; Fig. 1j), with many BAG2 condensates of ~1 μm. Other stressors such as temperature (42 °C), LPS, oligomycin (Supplementary Fig. 2) and MG132 (see below) also increase endogenous BAG2 condensation.

**BAG2 condensates are distinct from stress granules (SGs) and processing bodies (PBs).** The first clue that BAG2 condensates were distinct from SGs was the finding that sodium arsenite (NaAsO₂), among the different stressors we tried, was unique in that it neither induced BAG2 condensates nor significant co-localization of BAG2 with G3BP1 in stress granules (Supplementary Fig. 3a). In fact, a time course experiment in cells stably expressing clover-BAG2 and exposed to arsenite (500 μM, 120 min) (Fig. 2a) reduced the total area of BAG2 condensation by $2.3 \pm 0.3$-fold ($p = 1.1 \times 10^{-3}$, 120 min; Fig. 2b, Supplementary Movie 3). NaAsO₂ also decreased the percentage of clover-BAG2 cells that contained condensates from $46.4 \pm 10.0 \%$ to $4.4 \pm 3.4\%$ ($p = 2.0 \times 10^{-4}$; Fig. 2c). Arsenite treatment decreased BAG2 by western blot (Supplementary Fig. 3b). BAG2 condensation shows a preferential response to hyperosmotic stress over oxidative stress with arsenite[14]. To support this mechanistic distinction, we immunolabeled endogenous G3BP1, a marker for SGs, in clover-BAG2 cells after hyperosmotic stress. Although hyperosmotic stress can induce stress granules[1,15,16], most of the BAG2 signal (~80%) failed to co-localize with G3BP1 after hyperosmotic stress (Fig. 2d). Nor did pEGP-TIA1, another SG marker, colocalize with ruby-BAG2 (Supplementary Fig. 3c). Similarly, less than 20% of PBs colocalized with BAG2 condensates (Fig. 2e).

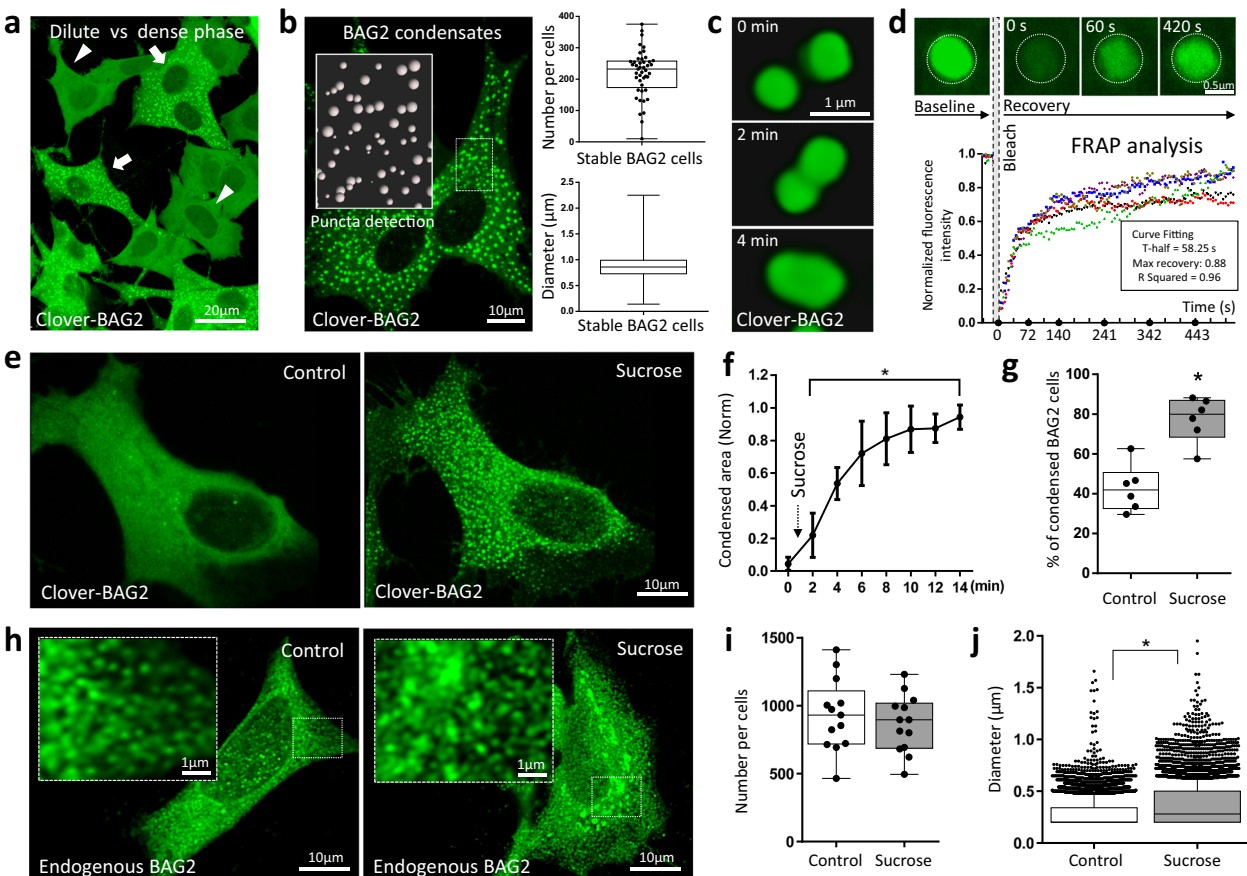

**Fig. 1 BAG2 condensation in response to stress. a** Representative image of cells expressing BAG2 in both dilute (arrowhead) and dense phases (arrow) in SH-SY5Y cells stably expressing clover-BAG2. 49.7 ± 12.5% of the clover-BAG2 population contained condensates (915 cells, $n = 5$, mean ± SD). **b** Representative image of cells containing BAG2 condensates illustrating puncta detection (insert). Number of condensates per cell (45 cells, $n = 3$); condensate diameter (2798 condensates, 10 cells, $n = 3$). Box: 25th and 75th percentiles. Line: median. Whiskers: smallest to the largest values. Dots: individual cell. **c** Image over time (Supplementary Movie 1): fusion of two BAG2 condensates. **d** Condensates dynamically exchanged BAG2 molecules with the dilute phase as demonstrated by FRAP analysis. Images showing representation of pre-bleaching (baseline), bleaching and post-bleaching fluorescence recovery; graph: fluorescence recovery quantification over time. Colors: different condensates. Insert: Fitting curve mean values for exponential recovery. **e** Representative image of a clover-BAG2 stable cell before and after stress (Sucrose 125 mM, 15 min). **f** Quantification of the total BAG2 condensation area over time. Condensation area is shown as the total sum of clover-BAG2 area normalized by its highest value, * from $p = 3 \times 10^{-2}$ (2 min) to $p < 1.0 \times 10^{-4}$ (14 min); one-way ANOVA followed by the Dunnett's test, $n = 3$, 8 cells, mean ± SD. **g** Percentage of clover-BAG2 cells that contained condensates before and after stress (Sucrose, 125 mM) ($n = 6$; 1023 cells for control and 1216 for stress, *$p = 4.0 \times 10^{-4}$, two-sided Student's $t$ test. Box and whiskers: same as in "**b**". Dots: replicates. Experiments in **a–g** performed in clover-BAG2 stable SH-SY5Y cells. **h** Immunostaining images of endogenous BAG2 in SH-SY5Y cells before and after stress (Sucrose 125 mM, 30 min). **i** Average number of 932.2 ± 264.4 BAG2 puncta/cell after exposure to stress, mean ± SD. Box and whiskers: same as in "**b**". 13 cells, $n = 3$. **J** BAG2 puncta diameter (*$p < 1.0 \times 10^{-4}$, two-sided Students' $t$ test, 12 cells, 10,883 puncta for control and 13 cells, 11,202 puncta for stress). Box: same as in "**b**". Whiskers: 10th and 90th percentile. Dots (individual condensate) below and above the whiskers are shown. $n$ = independent replicates.

RNA is a critical component of SGs and PBs[17]. mRuby-BAG2 stable cells were incubated, in vivo, with the cell-permeant SYTO 14 green fluorescent nucleic acid stain. In contrast to SGs and PBs, only a small percentage of BAG2 condensates colocalized with RNA (Fig. 2f). Taken together these observations suggest that BAG2 condensates represent a distinct membraneless organelle.

**Structural determinants of BAG2 condensates**. BAG2 can undergo multi-oligomerization in vitro[6] which depends on both the amino terminal coiled-coil domain and the carboxy terminal BAG2 domain. BAG2 oligomerization (Fig. 3a) likely lowers the energy barriers for LLPS[18] due to heterogeneous nucleation on the surface of the oligomer complexes. Based upon classical nucleation theory, heterogeneous LLPS lowers the free-energetic barrier for nucleation relative to homogeneous LLPS due to

increased multivalent and heterotypic interactions in presence of the oligomers. To investigate this possibility, two different clover-BAG2 mutants were initially cloned: BAG2-I160A (point mutation in the BAG2 domain) and BAG2-Δ20-61 (coiled-coil region deleted) (Fig. 3b). The point mutation disrupted both the interaction with Hsp70 and Hsp70 NEF activity, in vitro (Supplementary Fig. 4a). To determine whether these BAG2 mutations affected their ability to undergo condensation, clover-BAG2 wild type and the mutants (BAG2-I160A and BAG2-Δ20-61) were separately transfected into SH-SY5Y cells and subjected to hyperosmotic stress (Fig. 3c, images). In clover-BAG2 wild type cells, the percentage of clover-BAG2 cells that contained condensates increased from 12.0 ± 6.4% to 78.0 ± 11.1% ($p = 1.9 \times 10^{-7}$; Fig. 3c, graph). However, neither of the mutants induced BAG2 condensation before or after stress (Fig. 3c, images and graph). This finding suggested that BAG2 condensation

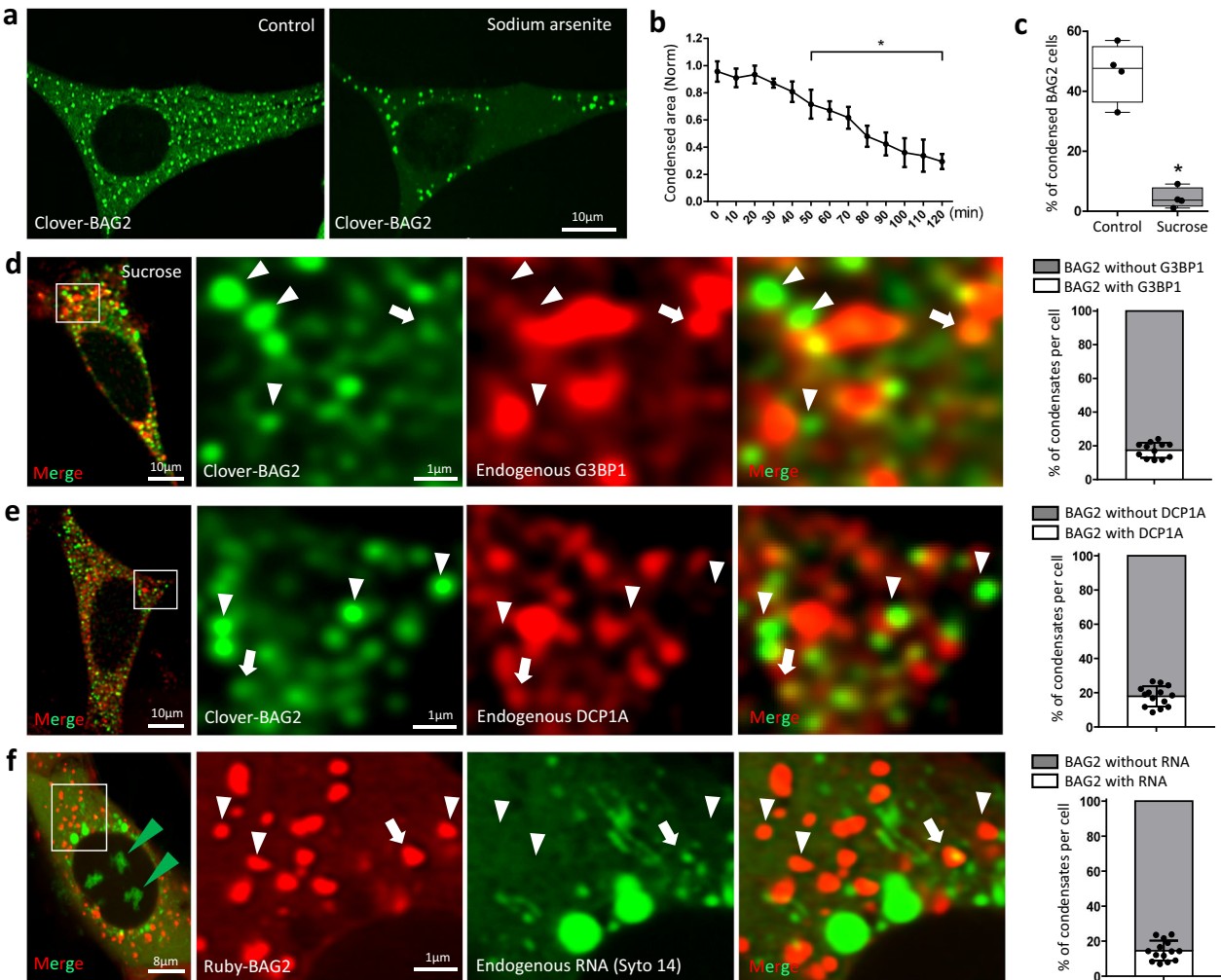

**Fig. 2 BAG2 condensates are distinct from stress granules and processing bodies.** Representative images (**a**) and BAG2 condensation area over time (**b**) in stably expressing clover-BAG2 SH-SY5Y cells exposed to arsenite (500 μM, 120 min, 3 independent replicates, * from $p = 4.8 \times 10^{-2}$ (50 min) to $p = 1.1 \times 10^{-3}$ (120 min); one-way ANOVA followed by the Dunnett's test, mean ± SD. **c** Percentage of clover-BAG2-positive cells that contained condensates compared to total clover-BAG2 cells (839 cells analyzed for the control and 1012 for arsenite in 4 independent replicates, *$p = 2.0 \times 10^{-4}$, two-sided Student's *t* test). Box and whiskers: the boxplots are centered around the median and extend from the 25th to 75th percentiles. The whiskers go down to the smallest value and up to the largest. Dots represent replicates. **d** Representative images and plot showing the immunostaining and quantification of co-localization between endogenous G3BP1, a stress granules marker, and BAG2 condensates in stably expressing clover-BAG2 SH-SY5Y cells exposed to sucrose (125 mM, 30 min; 1896 condensates analyzed in 12 cells from 3 independent replicates). **e** Representative images and plot showing the immunostaining and quantification of co-localization of endogenous DCP1A, a P-body marker, and BAG2 condensates in stably expressing clover-BAG2 SH-SY5Y (3621 BAG2 condensates analyzed in 14 cells from 3 independent replicates). **f** In vivo imaging of SH-SY5Y cells stably expressing ruby-BAG2 and incubated with the cell-permeant SYTO 14 green fluorescent RNA label. The percentage of BAG2 condensates co-localized with SYTO 14 is shown (1735 BAG2 condensates in 14 cells from 3 independent replicates). **d**–**f** Dots in the plots represent the percentage of BAG2 condensates colocalized with G3BP1, DCP1A and RNA (arrows in images) from each individual cell. Line is centered around the mean ± SD. Arrowheads (images): non-colocalized.

depends both on the BAG2 domain (BAG2-I160A) and the coiled-coil domain (BAG2-Δ20-61).

The phosphoserine at position 20 in the BAG2 sequence (the first amino acid of the coiled-coil domain deleted in BAG2-Δ20-61) has been described as a possible target of the p38 stress MAPK pathway[19,20]. To determine whether this site could mediate BAG2 condensation, a clover-BAG2 phosphomimetic (BAG2-S20E) and non-phosphorylatable (BAG2-S20A) analog (Fig. 3b, white arrow) were transiently expressed in SH-SY5Y cells and subjected to hyperosmotic stress (Fig. 3d, images). The BAG2 phosphomimetic mutant increased condensation under stress from $1.8 \pm 2.1\%$ to $49.8 \pm 8.5\%$ ($p = 3.5 \times 10^{-5}$; Fig. 3d, graphs), whereas the non-phosphorylatable analog failed to

induce BAG2 condensation. This result suggested a role for phosphorylation at serine 20 in BAG2 phase separation.

**BAG2 condensates protect cells against stress.** To identify a possible protective role of BAG2 condensation in the context of stress, clover-BAG2 wild type and the previously identified mutants (BAG2-Δ20-61 and BAG2-I160A) that did not undergo condensation during hyperosmotic stress were separately transfected into SH-SY5Y cells and subjected to an ATP based viability assay (CellTiter-Glo Luminescent Cell Viability Assay Kit, Promega). Hyperosmotic stress did not change cell viability in BAG2 wild type cells; however, it was decreased by $12.5 \pm 3.0\%$ in

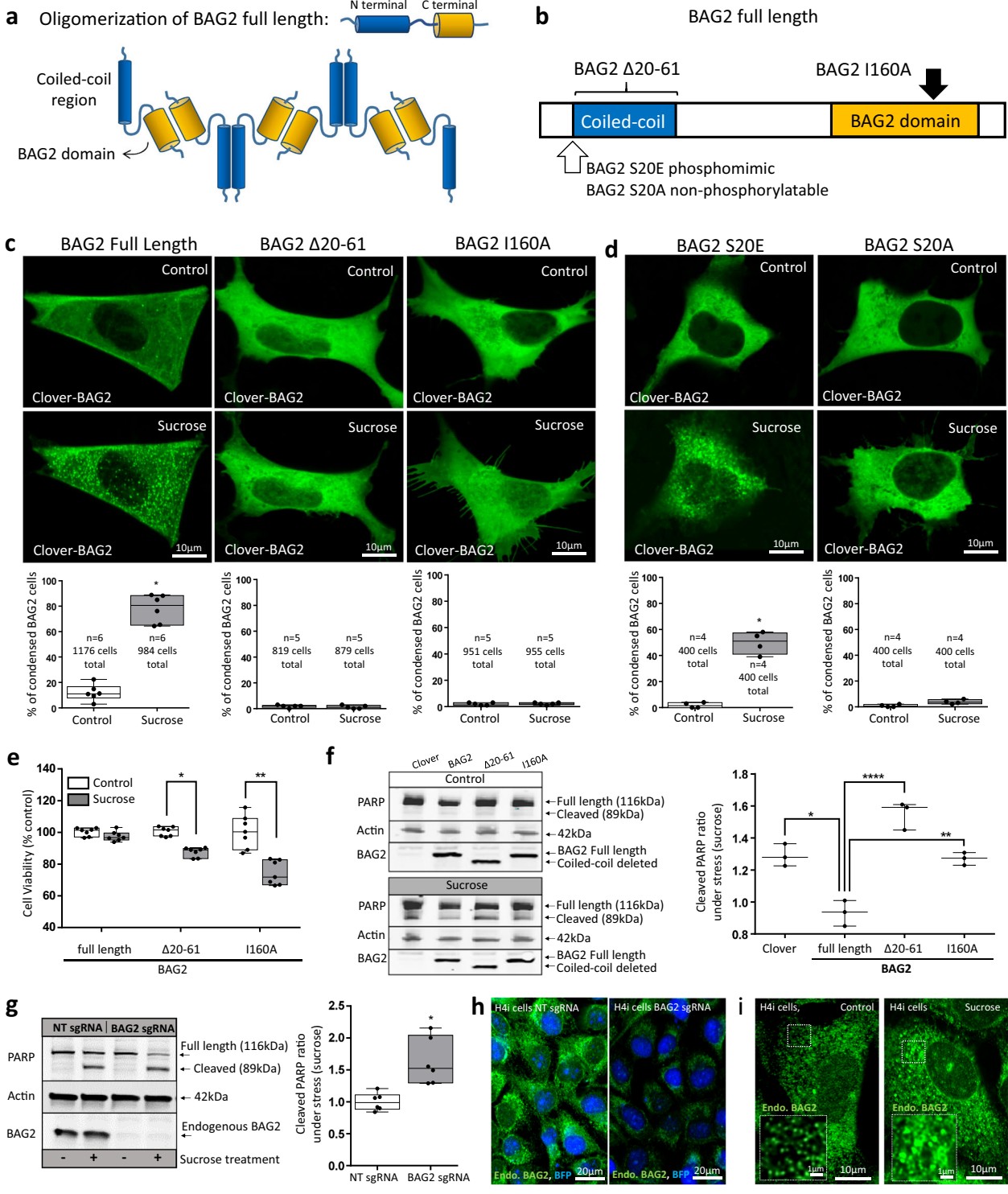

BAG2-$\Delta$20-61 cells ($p = 2.1 \times 10^{-3}$) and by $25.6 \pm 7.6\%$ in BAG2-I160A cells ($p = 7.5 \times 10^{-9}$, Fig. 3e) possibly due to the inhibition of BAG2 condensation.

The potentially protective effects of BAG2 condensates were further investigated in SH-SY5Y cells with a PARP antibody, a marker of cell death, after transfection of clover alone, clover-BAG2 wild type and clover-BAG2 mutants (BAG2-$\Delta$20-61 and BAG2-I160A). After 2 h of hyperosmotic stress, the levels of cleaved PARP ratio were decreased $38.7 \pm 7.5\%$ in clover-BAG2 wild type cells compared to clover alone ($p = 1.3 \times 10^{-3}$); whereas

cleaved PARP was increased $66.7 \pm 9.4\%$ in BAG2-$\Delta$20-61 cells ($p = 2.6 \times 10^{-5}$) and increased $36.6 \pm 4.2\%$ in BAG2-I160A cells ($p = 1.8 \times 10^{-3}$) as compared to wild type BAG2 cells (Fig. 3f). Similar results were also observed in cells stably expressing BAG2-$\Delta$20-61 and BAG2-I160A followed by hyperosmotic stress and probed by PARP antibody (Supplementary Fig. 4b).

Finally, the protective effect of endogenous BAG2 condensates was demonstrated by the effects of BAG2 repression. H4 cells constitutively expressing CRISPRi machinery[21] (H4i) were transduced with BAG2 sgRNA, selected, exposed to stress

**Fig. 3 Mutational analysis of BAG2 condensate function. a** Model of a multi-oligomerization structure of BAG2-full length protein representing the N and C termini interactions. **b** Mutation sites used to study BAG2 condensation in **c–f**. **c** Representative images (top) and plot (bottom) showing SH-SY5Y cells transfected with clover-BAG2 full length, clover-BAG2-Δ20-61 and clover-BAG2-I160A before and after stress (Sucrose 125 mM, 15 min). Plots: the percentage of clover-BAG2 cells that contained condensates before and after stress (*$p = 1.9 \times 10^{-7}$, two-sided Student's $t$ test). **d** Same experiment performed in **c** using clover-BAG2-S20E (phosphomimetic) and clover-BAG2-S20A (non-phosphorylatable) before and after stress (Sucrose 125 mM, 15 min, *$p = 3.5 \times 10^{-5}$, two-sided Student's $t$ test). **e** Cell viability as measured by the CellTiter-Glo(R) Luminescent Cell Viability Assay kit in control and after stress (500 mM, 2 h) in SH-SY5Y cells transfected with clover-BAG2 full length, clover-BAG2-Δ20-61 and clover-BAG2-I160A (7 independent replicates, *$p = 2.1 \times 10^{-3}$, **$p = 7.5 \times 10^{-9}$, two-way ANOVA followed by the Tukey`s test). **f** Western blotting against PARP, Actin and BAG2 in SH-SY5Y lysates of cells transfected with empty vector (Clover), BAG2 full length, BAG2-Δ20-61 and BAG2-I160A and subjected to stress (Sucrose 500 mM, 2 h). Blots and densitometry quantification of 3 independent replicates. Beta-actin was used as a loading control (*$p = 1.3 \times 10^{-3}$ for BAG2 full length; ****$p = 2.6 \times 10^{-5}$ for BAG2-Δ20-61; **$p = 1.8 \times 10^{-3}$ for BAG2-I160A; one-way ANOVA followed by the Tukey`s test). Line: median. Plots: smallest to the largest values. Points represent replicates. **g** Western blotting against PARP, Actin and BAG2 in H4i cells transduced with BAG2 sgRNA and NT sgRNA before and after stress (sucrose, 500 mM, 2 h). Six independent replicates *$p = 0.0027$; two-sided Student's $t$ test. **h** BAG2 immunocytochemistry images of H4i cells transduced with NT sgRNA and BAG2 sgRNA. BFP: CRISPRi machinery fluorescence marker (H4i). **i** BAG2 immunocytochemistry images of the regular H4 cells before and after stress (Sucrose, 125 mM, 30 min). **c–e**, **g** Box and whiskers: the boxplots are centered around the median and extend from the 25th to 75th percentiles. The whiskers go down to the smallest value and up to the largest. Dots represent replicates.

(sucrose) and probed with a PARP antibody. After 2 h of hyperosmotic stress, the levels of cleaved PARP ratio were increased $66.9 \pm 4.7\%$ in BAG2 sgRNA as compared to non-targeting sgRNA (NT or scramble shRNA) controls (Fig. 3g - graph). The knockdown efficiency was confirmed by WB (Fig. 3h - blots) and by ICC (Fig. 3h). Endogenous BAG2 condensates are also present in H4 cells and increase after sucrose treatment (Fig. 3i).

**BAG2 condensates are multicomponent entities**. To demonstrate the heterotypic multicomponent nature of BAG2 condensates we demonstrated a non-fixed saturation concentration of the dense phase both before and after hyperosmotic stress (Fig. 4a, b). Fluorescence correlation spectroscopy (FCS) was used to determine absolute concentrations of clover-BAG2 in the dilute and dense phases[22–24] as a means of measuring the transfer free energies of condensates before and after stress (Supplementary Fig. 5a, b). In a system composed of completely homotypic BAG2:BAG2 interactions, changes in the total cellular concentrations of BAG2 should only affect the volume ratio of BAG2 in the dilute and dense phase, while leaving the concentrations of the two phases unperturbed. Additionally, both the partition coefficient (1),

$$K = \frac{[Bag2]^{Dense}}{[Bag2]^{Dilute}}, \quad (1)$$

and the transfer free energy (2),

$$\triangle G^{tr} = -RT\ln K, \quad (2)$$

should be unperturbed across a range of BAG2 concentrations if only homotypic interactions were present[25]. In cells containing condensed BAG2, as the concentration of $[Bag2]^{Dilute}$ increased, so did the $\Delta G^{tr}$ both before and after hyperosmotic stress treatment (Fig. 4a) which is indicative of heterotypic interactions[25]. Further, when fitting individual FCS traces fluctuations in the measured diffusion coefficients were uncorrelated with measured concentrations, supporting the hypothesis that heterotypic interactions are playing an important role in this system (Supplementary Fig. 5c). These findings provided direct evidence that BAG2 condensates were driven by heterotypic interactions with other biomolecular components as active participants in the regulation of BAG2 phase transition.

Based upon the condensation of BAG2 with stress we determined whether BAG2 heterotypic interactions were modulated by cellular stress, as seen for other canonical LLPS systems such as stress granules[26]. By calculating the $\Delta G^{tr}$ of individual cells before and after hyperosmotic stress, changes in the interactions leading to BAG2 condensates were assessed (Fig. 4b). After a 15 min incubation with sucrose, the average $\Delta G^{tr}$ decreased by 0.17 kcal mole$^{-1}$ ($p = 1.3 \times 10^{-8}$). The more negative transfer free energy suggests that heterotypic interactions in the BAG2 condensates have increased in stressed cells. The more negative $\Delta G^{tr}$ after stress may be due to post-translational modifications on BAG2 (and other proteins) and/or flux of newly available interactions partners that provide the perturbation necessary to make phase transition more favorable.

BAG2 binds to the ATPase domain of the stress response heat shock protein 70 (Hsp70), thereby driving client proteins to the proteasome[9]. To identify the components within BAG2 condensates, SH-SY5Y cells were transfected with pEYFP-BAG2 and DsRed-Hsp70, subjected to hyperosmotic stress, and imaged by time lapse. In this experiment, Hsp70 demixed with BAG2 condensates 5 min after stress (Fig. 4c). In clover-BAG2 stable cells, endogenous Hsp70 labeled the BAG2 condensates with $80.7 \pm 9.8\%$ colocalization after 30 min of hyperosmotic stress (Fig. 4d). We also demonstrated that BAG2 condensates contain the 20S proteasome and not the 26S. When clover-BAG2 stably transfected cells were subjected to hyperosmotic stress (30 min) and stained with 20S alpha 5 proteasome antibody, $60.2 \pm 9.4\%$ colocalization was observed with BAG2 condensates (Fig. 4e). Similar results were obtained in COS-7 cells (Supplementary Fig. 6a). In the stably transfected SH-SY5Y cells, BAG2 condensates also label with PA28 gamma (~80%; Fig. 4f) and PA28 beta (~60%) after hyperosmotic stress (Supplementary Fig. 6b). The PA28 proteins (also known as 11S, REG or PMSE) can markedly enhance degradation rates by the 20S proteasome[27]. Although PA28 is described as localized to the nucleus, after stress PA28 is readily detectable in the cytoplasm[28]. The detection of the alpha 5 subunit does not exclude the concomitant presence of the 26S. Endogenous BAG2 condensates show only a very small percent colocalization with the 19S cap (PA700), which is associated with 26S (Supplementary Fig. 6c). However, as noted in the transfection experiments, a large percentage of PA28 gamma is associated with endogenous BAG2 condensates after hyperosmotic stress (Supplementary Fig. 6d) suggesting that the 20S proteasome is the major component of BAG2 condensates. In summary, the composition of BAG2 condensates define a distinct cellular organelle which highly co-localized with Hsp70 and the activated 20S proteasome while far less frequently co-localizing with RNA, G3BP1, DCP1A and 19S cap.

**BAG2 condensates mediate degradation**. Several lines of evidence including the absence of a ubiquitin-like domain (Ubl) on

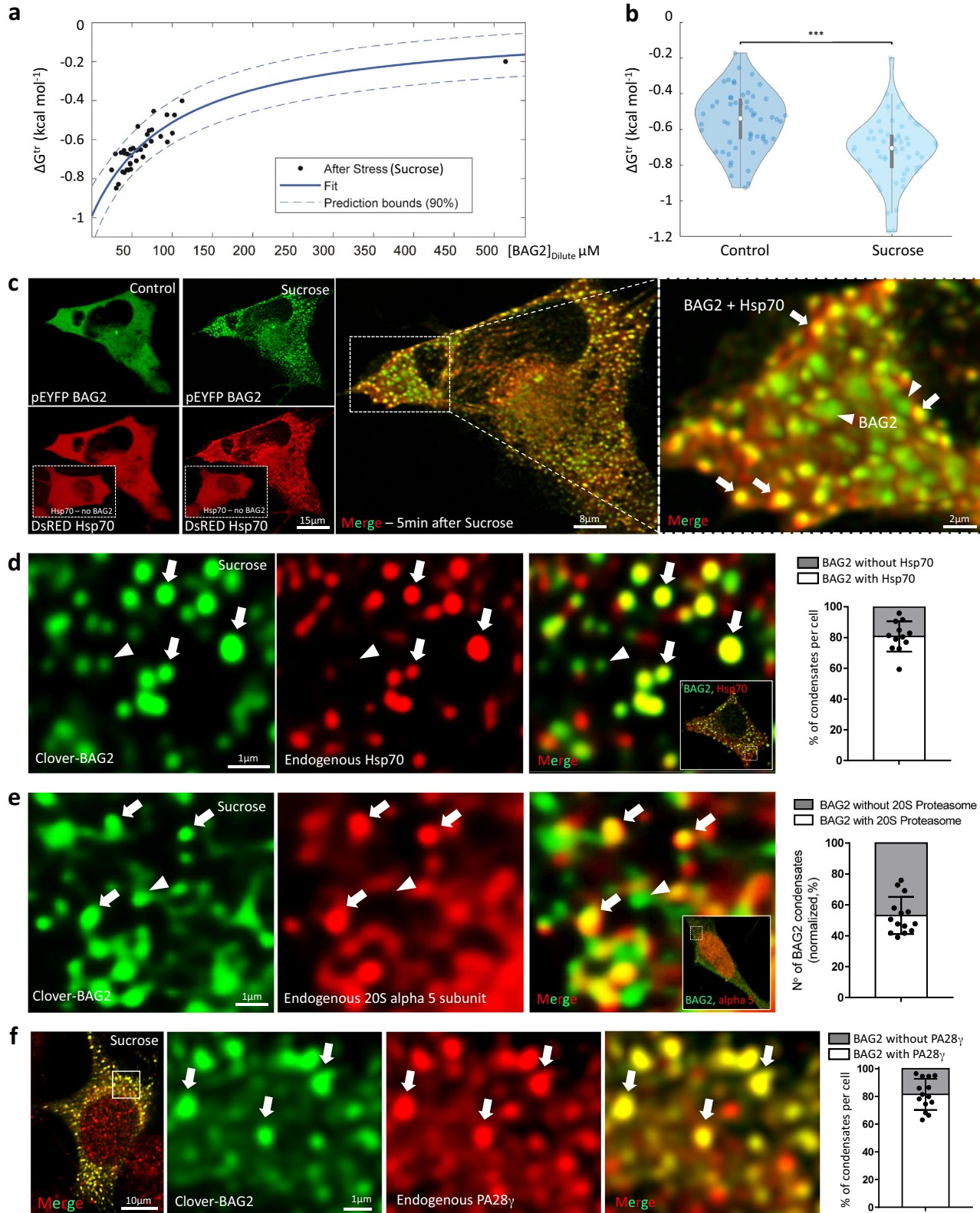

BAG2 molecule and its function as an inhibitor of the ubiquitin E3 ligase CHIP[7] suggest that BAG2 condensates degrade proteins independently of ubiquitin. To validate this hypothesis, clover-BAG2 stable cells were subjected to hyperosmotic stress (30 min) and stained for endogenous ubiquitin-K48 antibody[29]. Less than 20% of the BAG2 condensates were co-localized with ubiquitin-

K48 label after stress (Fig. 5a). Similar results after hyperosmotic stress, were observed using the UBCJ2 antibody that recognizes mono- and poly-ubiquitinylated conjugates (Supplementary Fig. 7). Furthermore, only a minority of the endogenous BAG2 condensates colocalized with CHIP (~20%) (Supplementary Fig. 8a). Interestingly, ~80% colocalization with CHIP was

**Fig. 4 BAG2 condensates are multicomponent entities. a** DeltaG Plot, dependence of the transfer free energy on the concentration of BAG2 in the dilute phase. The solid line represents the best fit of the data, while the dotted lines represent the 90% mean confidence interval (see methods). The nonlinear, increasing $\Delta G^{tr}$ as a function of BAG2 concentration indicates that heterotypic interactions control BAG2 phase separation. **b** SH-SY5Y cells stably expressing clover-BAG2 were analyzed by fluorescence correlation spectroscopy (FCS) before and after stress. Stress increased the heterotypic interactions inside the BAG2 condensates. Violin plots of the transfer free energy of condensates formation before and after the addition of sucrose. The two white circles represent the medians, while the gray bars are the interquartile range. Each dot represents the average transfer free energy of a single cell (5 independent replicates, 54 cells analyzed, two-sided Pairwise Student's $t$ test, ***$p = 4.6 \times 10^{-6}$). **c** SH-SH5Y cells transiently expressing pEYFP-BAG2 and DsRED-Hsp70 subjected to stress (Sucrose 125 mM, 5 min). Arrows in the zoomed right panel indicate co-localization of Hsp70 and BAG2 condensates. Arrowheads indicate BAG2 condensates without Hsp70. Inserts in the Hsp70 panels (red) show representative images of SH-SY5Y cells transfected only with Hsp70 (no BAG2) and subjected to stress highlighting the absence of Hsp70 condensates under this condition. Images of endogenous Hsp70 (**d**), endogenous 20S alpha 5 subunit of the proteosome (**e**) or endogenous PA28 gamma (**f**) immunostaining in SH-SY5Y cells stably expressing clover-BAG2 subjected to stress (Sucrose 125 mM, 30 min). Arrows indicate colocalization between BAG2 condensates and Hsp70 (**d**), proteosome 20S alpha 5 subunit (**e**) or PA28 gamma (**f**) while arrowheads highlight the presence of clover-BAG2 condensates with absence of Hsp70 or proteosome 20S alpha 5 subunit immunolabeling. The quantification of the co-localization is graphically depicted [4020 BAG2 condensates analyzed in 12 cells (**d**); 3996 BAG2 condensates in 14 cells (**e**) and 1988 BAG2 condensates in 14 cells)]. All from 3 independent replicates, **d**–**f**. Dots represent the percentage of BAG2 condensates colocalized with Hsp70, 20S alpha 5 subunit and PA28 gamma from each individual cell. Line is centered around the mean ± SD.

observed among the largest 5% BAG2 condensates (Supplementary Fig. 8a) suggesting a possible shift in degradation process within a bigger and crowded environmental. The absence of a role for ubiquitin in BAG2 condensates was additionally demonstrated in clover-BAG2 stable SH-SY5Y cells pre-treated with the ubiquitin E1 inhibitor MLN-7243, 1 μM, 1 h prior[2], and subjected to hyperosmotic stress. MLN-7243 did not block the formation of BAG condensates (Supplementary Fig. 8b) but blocked the high molecular weight species formed by ubiquitin-K48 targeting linkages after hyperosmotic stress (30 min) (Supplementary Fig. 8c) suggesting that BAG2 condensates do not utilize an E1 ubiquitin conjugate enzyme or ubiquitin within BAG2 condensates.

To determine directly whether BAG2 condensates mediated protein degradation, we used the client protein ZsGreen fused to a PEST sequence (pZsProSensor-1; Takara Bio Inc). The PEST sequence efficiently and rapidly drives ZsGreen degradation in a ubiquitin independent manner through the 20S proteasome[30]. SH-SY5Y and mRuby-BAG2 stable SH-SY5Y cells were both transiently transfected with ZsGreen alone (−PEST) or ZsGreen fused to PEST (+PEST). As expected, the +PEST signal was markedly decreased in both cell types as compared to its respective −PEST ($p = 3.5 \times 10^{-9}$ for control and $p = 8.9 \times 10^{-11}$ for stably BAG2 cells; Fig. 5b); however, the presence of +PEST had a higher clearance of ZsGreen in mRuby-BAG2 stable cells as compared to +PEST control SH-SY5Y cells ($p = 1.2 \times 10^{-2}$; Fig. 5b). Interestingly, in ruby-BAG2 stable cells the +PEST signal was co-localized with BAG2 condensates (Fig. 5c) in contrast to the −PEST signal that showed no co-localization with BAG2 condensates (Fig. 5d). These results support the hypothesis that BAG2 condensates degrade clients in a ubiquitin independent manner through the 20S proteasome.

**BAG2 condensates degrade Tau under stress conditions.** As a client of BAG2, Tau can undergo degradation independently of ubiquitin[9,31,32]; however, the role of condensates to Tau proteostasis is not well established[33]. To explore this role, SH-SY5Y cells were transiently transfected with clover-BAG2 and mRuby-Tau and the population of cells with both markers were analyzed. BAG2-containing cells fell into two clearly distinctive populations similar to the same experiments described above without Tau. A population with large numbers of BAG2 condensates, and a second population in which nearly all of the BAG2 was in the dilute phase (Fig. 6a, arrow vs arrowhead, green image). We quantified Tau intensity levels in these two populations. Cells with BAG2 condensates had $1.8 \pm 0.6$-fold increase in Tau levels compared to cells in which BAG2 was only present in the dilute

phase ($p = 2.4 \times 10^{-14}$; Fig. 6a, graph) suggesting that high levels of Tau preferentially recruit BAG2 condensates.

As observed previously[9], BAG2 aligned along the microtubules; however, in these experiments, we further report that BAG2 does so in the form of a condensate (Fig. 6a, insert; Fig. 6b; Supplementary Fig. 9a, b; Supplementary Movie 4). This observation suggests that Tau, while in proximity to the microtubule, is associated with a condensate implying that when Tau is associated with condensates it is unlikely to be bound to the microtubules. Parallel results were also observed in COS7 cells (Supplementary Fig. 9c). Support for attraction of the unbound form of Tau to condensates derived from observations after treatment with vinblastine (1 h), a microtubule depolymerizing agent. This treatment markedly increased BAG2 condensation (Supplementary Fig 9d, Supplementary Movie 5). In the next experiments, clover-BAG2 and mRuby-Tau were stably expressed in SH-SY5Y cells encoded by clover-BAG2-P2A-mRuby-Tau joined by the self-cleaving peptide P2A[34] so that Tau and BAG2 localize independently after their translation. In the absence of stress, BAG2 was mainly observed in the dilute phase with few condensates (Fig. 6c, left images). Hyperosmotic stress (15 min) rapidly induced BAG2 condensates that aligned along microtubules (Fig. 6c, right images). Similarly, endogenous BAG2 and Tau also increased in their association following hyperosmotic stress (Supplementary Fig 10a).

To capture Tau and BAG2 interactions within BAG2 condensates we performed FLIM-FRET measurements using the clover-mRuby2 FRET pair (Fig. 6d). After hyperosmotic stress, the whole cell interaction lifetime dropped compared to unstressed cells, which have few BAG2 condensates, but the decrease was not significant (from $3.15 \pm 0.07$ ns to $3.03 \pm 0.09$ ns, Fig. 6d). However, by segmenting the images and fitting the subset of pixels that belonged exclusively to BAG2 condensates, the measured lifetime was $2.8 \pm 0.1$ ns, significantly lower than whole control cells ($p = 2.7 \times 10^{-5}$) or whole cells after stress ($p = 2.8 \times 10^{-2}$) (Fig. 6d). The value of 2.8 ns is in good agreement with previous observations made for FRET from fused clover-mRuby constructs[35]. When measuring only the dilute BAG2 phase in stressed cells (excluding condensates) the lifetime also differed significantly from condensates ($3.1 \pm 0.1$ ns, $p = 2.6 \times 10^{-3}$) (Fig. 6d), a value very close to that of free clover ($3.15 \pm 0.07$ ns) indicating that no FRET occurred. Together these data suggest that the majority of BAG2-Tau interactions occur within condensates.

Finally, we sought direct detection of increased Tau degradation by BAG2 under hyperosmotic stress. SH-SY5Y cells

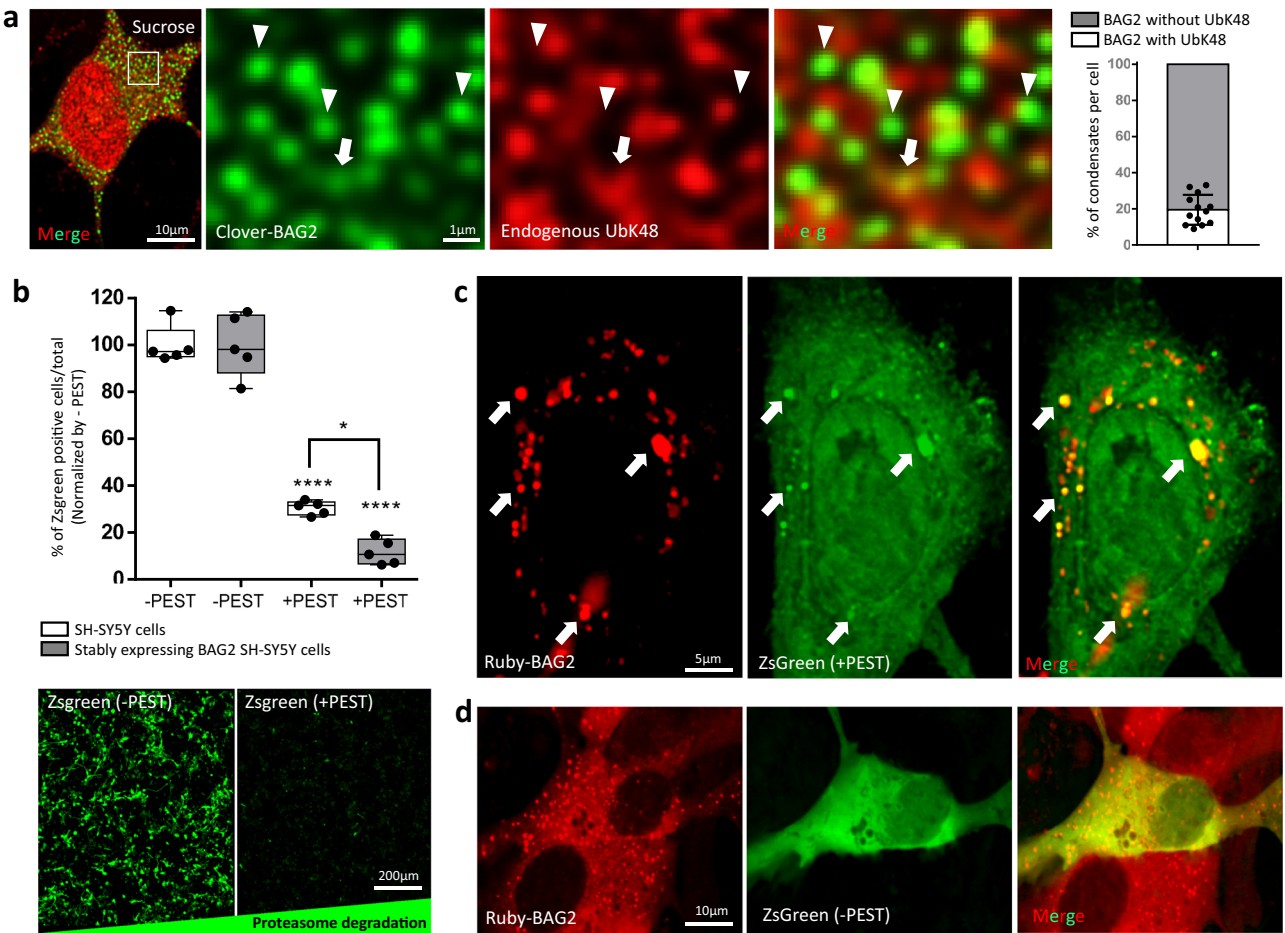

**Fig. 5 BAG2 condensates function as degradation granules. a** SH-SY5Y cells stably expressing clover-BAG2 immunostained for endogenous ubiquitin-K48. Arrows indicate colocalization between clover-BAG2 and ubiquitin-K48. In most BAG2 condensates ubiquitin-K48 immunolabeling was absent (arrowhead). The quantification of the colocalization is shown (2752 BAG2 condensates analyzed in 13 cells from 3 independent replicates). Dots represent the percentage of BAG2 condensates colocalized with ubiquitin-K48 from each individual cell. Line is centered around the mean ± SD. **b** Percentage of ZsGreen positive cell per total in SH-SY5Y cells and ruby-BAG2 stable cells transiently expressing ZsGreen alone (−PEST) or ZsGreen fused to PEST (+PEST). As expected, the +PEST signal was markedly decreased in both cell types as compared to its respective −PEST (****$p = 3.5 \times 10^{-9}$ for control and ****$p = 8.9 \times 10^{-11}$ for stably BAG2 cells); however, the presence of +PEST had a higher clearance of ZsGreen in mRuby-BAG2 stable cells as compared to +PEST control SH-SY5Y cells (*$p = 1.2 \times 10^{-2}$). Two-way ANOVA followed by the Tukey's test, 5 independent replicates. Representative images of the ZsGreen labeling are shown at the bottom. Box and whiskers: the boxplots are centered around the median and extend from the 25th to 75th percentiles. The whiskers go down to the smallest value and up to the largest. Dots represent replicates. **c** Representative images using higher laser intensity of ruby-BAG2 stable cells expressing the ZsGreen (+PEST) showed co-localization between +PEST signal and BAG2 condensates (arrows). 3 independent replicates. **d** As a control, ruby-BAG2 cells expressing the ZsGreen alone (−PEST) showed no co-localization between signals. 3 independent replicates.

expressing either Tau or Tau and BAG2 were subjected to 2 h of hyperosmotic stress and probed by western blotting with antibodies PHF-1, MC-1, AT-8 and Tau-5 (Fig. 6e, for blots see Supplementary Fig. 10b). Under stress, the presence of BAG2 resulted in $3.66 \pm 2.03$-fold decrease in PHF-1/Tau-5 ($p = 2.7 \times 10^{-3}$), $0.69 \pm 0.63$-fold decrease in MC-1/Tau5 ($p = 1.7 \times 10^{-2}$) and $1.13 \pm 0.90$-fold decrease in AT8/Tau5 ($p = 4.1 \times 10^{-2}$) (Fig. 6e). A key site of BAG2 action appears to be on the microtubule as after 2 h of hyperosmotic stress, BAG2 condensates remained associated with Tau on the microtubules (Supplementary Fig. 10c).

**LAMP-1, p62/SQSTM1 and BAG2 condensates**. BAG2 condensates did not co-localize with the lysosomal marker, LAMP-1 (Fig. 7a, insert) or p62/SQSTM1 (Fig. 7b, insert), a key protein involved in switching between the UPS and autophagy

pathways[36] mediated by binding to polyubiquitin[37,38]. Treatment with bafilomycin A$_1$ (BafA1), an autophagy inhibitor, had no effect on BAG2 condensates (Fig. 7c vs. a, b), suggesting that BAG2 condensates were not primarily associated with autophagy. However, in the context of proteasomal inhibition with MG132, autophagy is activated[39] and aggresomes form[40]. Exposure of SH-SY5Y cells to MG132 increased the percentage of cells with single large (>3 μm) endogenous BAG2-labeled juxtanuclear structures that impinged upon and distorted the contour of the nuclear envelope typical of aggresomes. These structures were present in $3.9 \pm 2.8\%$ of non-treated cells and $84.6 \pm 11.3\%$ in MG-132 treated cells ($p = 3.2 \times 10^{-10}$; Fig. 7d). As expected, these structures immune-labeled with LAMP-1 (Fig. 7e, Supplementary Fig. 11a) and p62/SQSTM1 (Fig. 7g, Supplementary Fig. 11b) suggesting that BAG2 condensates and autophagy related proteins migrate to this aggresome-like structure. LAMP-1 labeling predominated in regions devoid of BAG2 signal (Fig. 7e) with

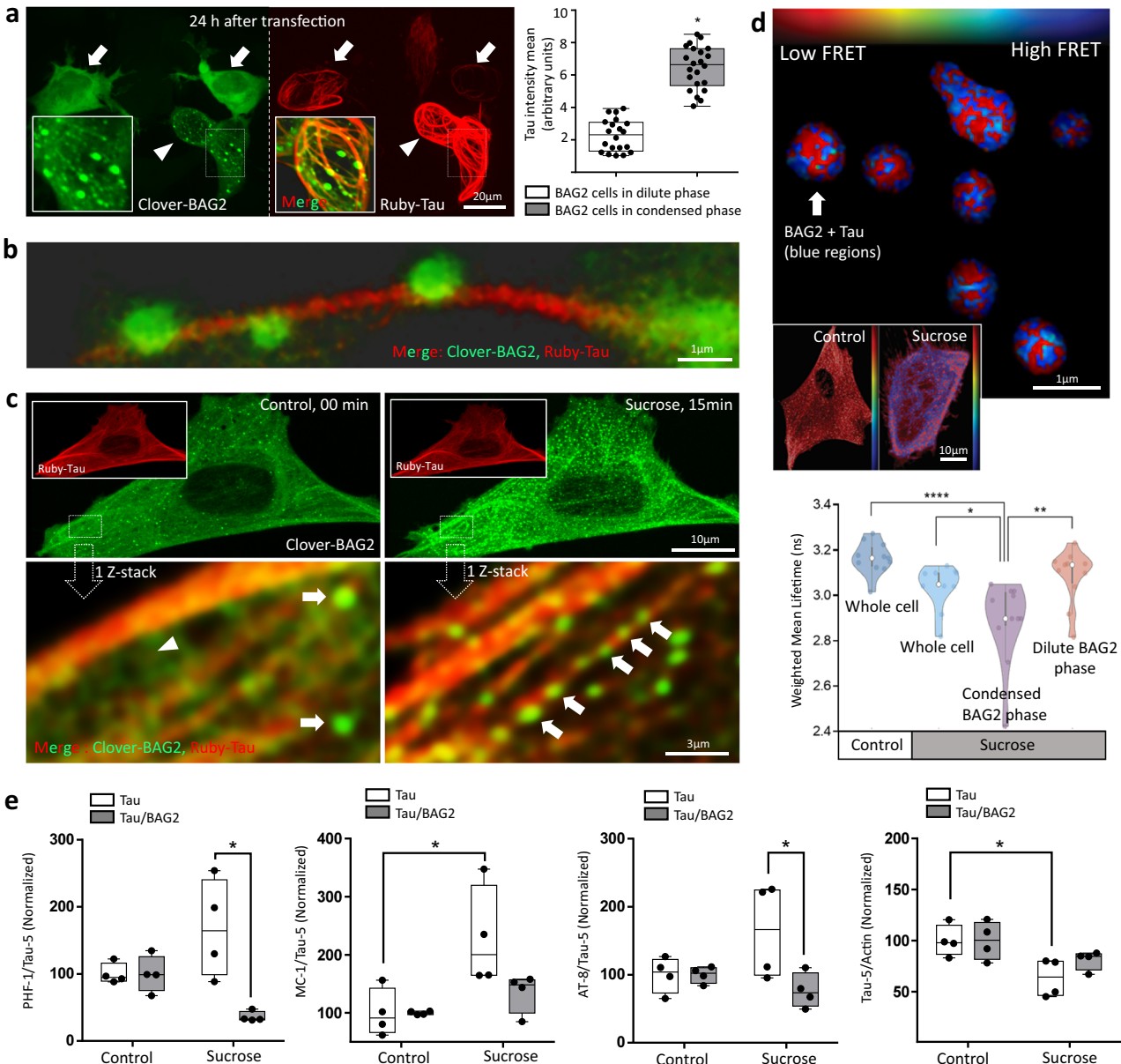

**Fig. 6 Function of BAG2 condensates. a** SH-SY5Y cells transiently expressing ruby-Tau and clover-BAG2 after 24 h show BAG2 in dilute phase (arrow) and dense phase (arrowhead). Fluorescence of Tau intensity was normalized by its highest value in these two populations ($n = 3$, 20 cells for dilute and 22 for dense phase, $*p = 2.4 \times 10^{-14}$, two-sided Student's $t$ test). Box: 25th and 75th percentiles. Line: median. Whiskers: smallest to the largest values. Dots: represent individual cell. **b** BAG2 condensates aligned along microtubules (Supplementary Movie 4). **c** SH-SY5Y stably expressing clover-BAG2 and ruby-Tau after stress (sucrose, 15 min) increase BAG2 condensates and their alignment along the microtubules (arrows). Arrowhead: BAG2 dilute phase. **d** BAG2-Tau interaction by FLIM-FRET. Image displays the fluorescence lifetime (red: 3.2 ns - no energy transfer; blue: lower lifetime - energy transfer). Graphical representation of the lifetime signal before and after stress in the whole cell, condensates or dilute phase. Over the entire cell, lifetimes did not change significantly after stress. However, the lifetime images indicated a significant FRET signal within condensates after stress (top). By segmenting the images and fitting pixels belonging to distinct condensates, the fit had an average weighted mean lifetime of 2.8 ± 0.1 ns, significantly lower compared to control ($****p = 2.7 \times 10^{-5}$) or to whole cells after stress ($*p = 2.8 \times 10^{-2}$). When measuring only the dilute BAG2 phase in stressed cells (excluding condensates) the average weighted mean lifetime was 3.1 ± 0.1 ns which also differed from the condensates ($**p = 2.6 \times 10^{-3}$). Therefore, the majority of BAG2-Tau interactions occur in condensates. 11 cells, $n = 3$; One-way ANOVA followed by the Tukey`s test. Floating lines: 25th and 75th percentiles. Circle: median. Violin: smallest to the largest values. Dots: individual cell. **e** Cells stably expressing Tau or Tau-P2A-BAG2 were subjected to stress (sucrose 500 mM, 2 h) and resolved with Tau-5, PHF-1, MC-1, and AT-8. $n = 4$ (see Supplementary Information). Under stress, the presence of BAG2 decreased the ratio of PHF ($*p = 2.6 \times 10^{-3}$), MC-1 ($*p = 1.7 \times 10^{-2}$) and AT-8 ($*p = 4.0 \times 10^{-2}$) to total Tau. two-way ANOVA followed by the Tukey`s test. Box and whiskers: same as in "**a**". Dots represent replicates.

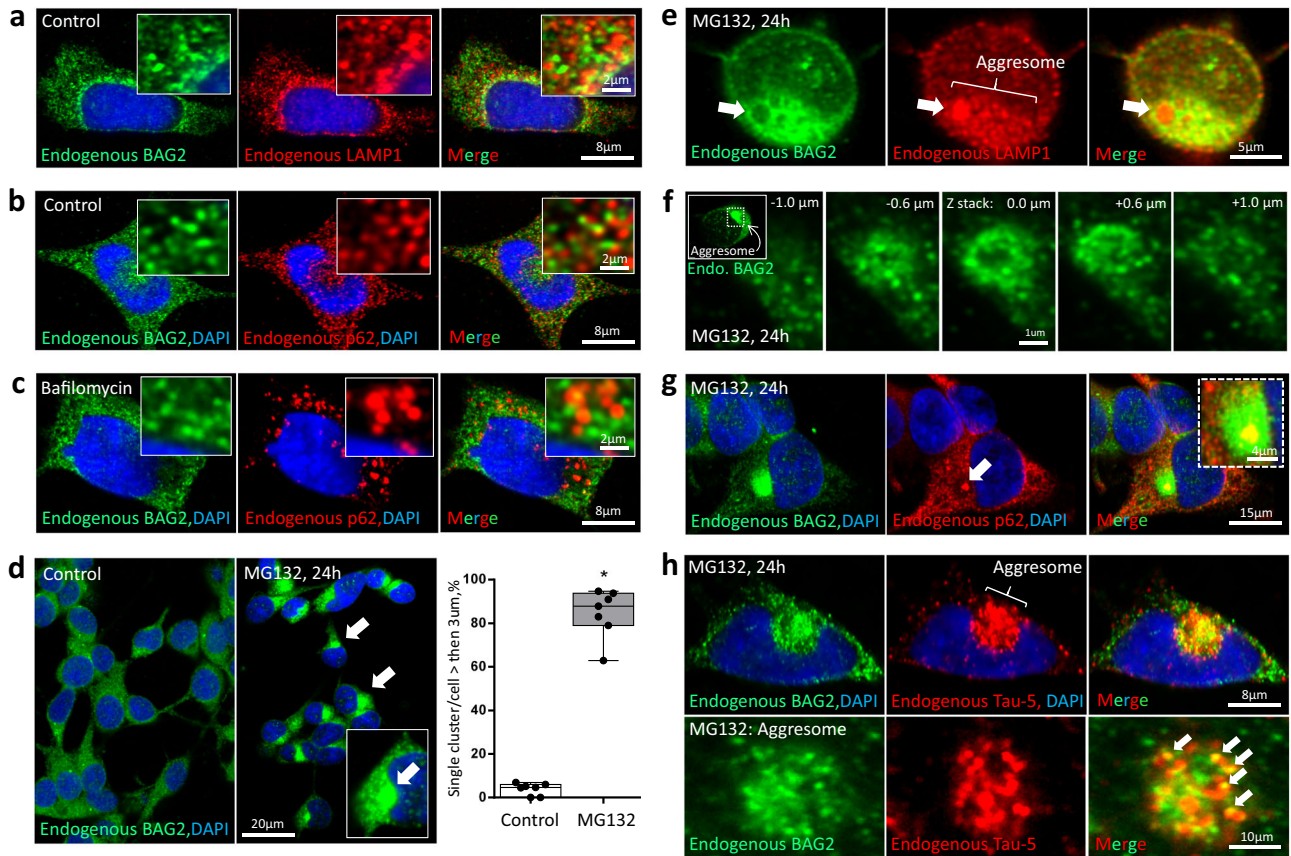

**Fig. 7 BAG2 condensates with UPS inhibition.** Co-immunostaining of BAG2 and LAMP-1 (**a**) or BAG2 and p62/SQSTM1 (**b**) in control cells, showing no co-localization between markers (insert). **c** Bafilomycin (15 h, 50 nM), an autophagy inhibitor, did not change the endogenous BAG2 pattern. **d** Immunostaining of endogenous BAG2 in SH-SY5Y cells treated with MG132 (24 h, 10 μM) showing a robust increase in the percentage of BAG2 cells containing a single aggresome-like structure (>3 μm) of endogenous BAG2 condensates, not observed in non-treated cells (273 cells, 7 independent replicates; *p = 3.2 × 10^{-10}, two-sided Student's *t* test). Box and whiskers: the boxplots are centered around the median and extend from the 25th to 75th percentiles. The whiskers go down to the smallest value and up to the largest. Dots represent replicates. **e** Co-immunostaining for endogenous BAG2 and LAMP-1 after 24 h of MG132 treatment revealed intense staining of both markers in the aggresome; however, LAMP-1 showed strong staining at sites devoid of BAG2 signal (arrow). **f** Images showing a higher magnification of 5 different Z-stack acquisition (from −1 μm to +1 μm) of the aggresome (see insert) revealing multiple small BAG2 condensates (~0.2 μm) surrounding a well-defined BAG2-negative core. **g** Co-immunostaining for endogenous BAG2 and p62/SQSTM1 after 24 h of MG132 revealed intense staining for both markers in the aggresome (see also Supplementary Fig 11b). p62/SQSTM1 forms a well-defined region (arrow) within the BAG2 signal (Insert). **h** SH-SY5Y cells treated with MG132 (24 h, 10 μM) accumulate endogenous Tau-5 in the perinuclear region (upper panel). In this region, some BAG2 condensates demixed with Tau (bottom panel, arrows). Experiments (**a–c** and **e–h**) were performed in 4 independent replicates.

multiple small BAG2 condensates (~0.2 μm) surrounding this BAG2 negative core (Fig. 7f). p62/SQSTM1 labels a well-defined region within the locale of the BAG2 signal (Fig. 7g - insert). This observation suggested that within the aggresome-like structure, clients appeared segregated according to their mechanism of protein degradation.

Tau was also present in the large aggresome-like perinuclear structures (Fig. 7h, upper panel, Supplementary Fig. 11c) after MG132 exposure. Some BAG2 condensates co-localized with Tau (Fig. 7h, bottom panel) consistent with its known ubiquitin-independent degradation mechanism[9,31,32]; whereas, Tau was not observed in regions devoid of BAG2 devoid (Supplementary Fig. 11d).

## Discussion

Stress driven condensation is associated with heterogeneous nucleation of myriad bio-macromolecules including RNA and ubiquitin as key components that provide multi-valent interactions to facilitate protein degradation and sequester RNA when polysomes undergo disassembly and thereby abrogate protein

translation. One means for implementing these processes is the formation of membraneless phase-separated organelles that concentrate and localize specialized molecular machinery such as stress granules, processing bodies[41], and RNA transport granules[42]. BAG2 condensates, although functionally linked to stress[43–45] and capable of mediating folding and degradation decisions does not involve RNA or ubiquitin and operates as a distinct system within the shared cytoplasmic dilute phase of membraneless organelles. BAG2 condensates, which have been described as puncta[9], are shown here to meet the commonly accepted criteria for a phase-separated entity: spherical structures that fuse and recover from photobleaching. However, these criteria do not definitively prove LLPS. Although 1,6-hexanediol has emerged as an indicator of LLPS, it was not used here because it can change the permeability of membranes and thus lead to artifacts[46]. We provided additional evidence for BAG2 phase separation in cells by showing a change in the partition coefficient expected with a multi-component condensate and genetic manipulations that changed the valency and, in so doing, completely obliterated hyperosmotic stress-induced BAG2

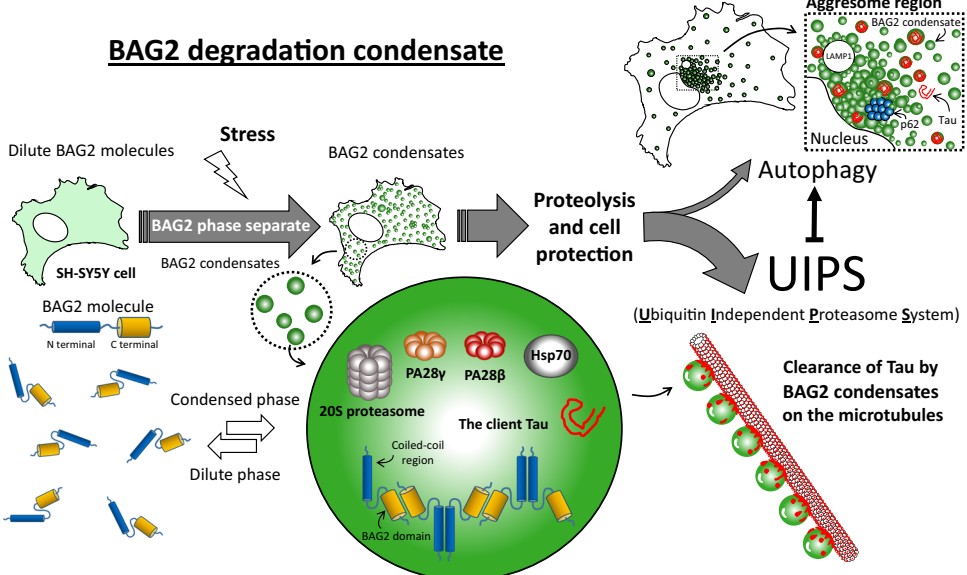

**Fig. 8 Schematic representation of the BAG2 degradation condensate.** BAG2 marks a distinct phase-separated membraneless organelle, triggered by several forms of stress, particularly hyper-osmotic stress. BAG2 condensation depends on both the amino terminal coiled-coil domain and the carboxy terminal BAG2 domain. Distinct from well-known condensates such as stress granules and processing bodies, BAG2-containing granules lack RNA, lack ubiquitin and promote client degradation in a ubiquitin-independent manner via the 20S proteasome (UIPS). Components of these condensates include the chaperone HSP-70 and the 20S proteasome activated by members of the PA28 (PMSE) family. These organelles protect the viability of cells from stress and are capable of trafficking directly to their clients, in the case of Tau, on the microtubule. BAG2 condensates did not co-localize with LAMP-1 or p62-SQSTM1. When the proteasome is inhibited, BAG2 condensates can traffic to an aggresome-like structure with the autophagy markers LAMP-1 and p62/SQSTM1. LAMP-1 signal is predominantly in regions devoid of BAG2 and p62/SQSTM1 antibody labels a discrete region in proximity to the BAG2 signal. This observation suggested that within the aggresome-like structure, clients were possibly segregated according to their mechanism of protein degradation.

condensation. As was the case for BAG2 condensates (Fig. 3b), previous studies have shown that phase separation in some proteins depends upon a coiled-coil domain as observed for ASK3 under hyperosmotic stress[47] or ORF1 RNA chaperone[48]. Interestingly, the BAG2 phosphorylation site at the first amino acid of the coiled-coil domain, modulates BAG2 phase separation (Fig. 3d).

Of interest is the ability of BAG2 condensates to assemble with its degradation or refolding machinery at the cellular locale of its client both in the case of Tau and more generally with ZsGreen fused to the PEST sequence. Condensation is more favorable to form on the microtubules due to the increment of heterogeneous driven nucleation with a lower free-energy barrier. BAG2 condensates appear to provide a safe reservoir for Tau off the microtubule. BAG2 condensates deliver clients directly to the 20S proteasome, which partitions with the dense phase and, in contrast to the 26S proteasome, does so independently of ubiquitination. The affinity of natively folded proteins for the 26S proteasome is extremely low, and therefore it is necessary to increase the affinity by hydrolysis of the protein with a polyubiquitin chain that requires a series of enzymatic steps carried out by three classes of enzymes (E1, E2, and E3)[49]. In contrast, the 20S when capped by ATP- and ubiquitin-independent PA28 activator, which is present in BAG2 condensates, profoundly increases its proteolytic activity[50] and thereby creates a distinct set of appropriate candidate clients (Fig. 8). Interestingly, gene therapy with PA28 gamma in an animal model of Huntington's disease improved motor coordination[51] which suggests that the BAG2 condensate could serve as a therapeutic target[52].

Finally, Tau degradation pathways must be considered regarding the degradation mechanism intended to eliminate normal tau or pathological tau or some intermediate state as Tau acquires the post-translational modifications[53] that ultimately lead to its accumulation as a fibrillar inclusion. Misfolded proteins from Alzheimer's, Parkinson's, and Huntington's disease can potently impair the proteasome[54]. Pathogenic variants of Tau are targeted to lysosomes via chaperone-mediated autophagy and eventually block this autophagic pathway[55]. Tau delivery to the proteasome is likely to function in the normal Tau degradation pathway, but at some point in the pathogenesis of a tauopathy, autophagy subsumes a role. By forcing the system toward autophagy with MG132, and thereby inducing large p62-containing aggregates that mediate ubiquitin-dependent "aggrephagy"[36,56], BAG2 condensates engage in cross-talk with and traffic toward an aggresome-like structure (Fig. 8).

## Methods

**Antibodies, reagents, and plasmids.** The following antibodies we used for immunoblotting (WB) and/or immunofluorescence (IF): Invitrogen, rabbit anti-BAG2 polyclonal antibody [cat # PA5-96794, 1:1000 (WB) and 1:250 (IF)]; StressMarq Bioscience - mouse anti-human HSP70-HSC70 Monoclonal [cat # SMC-104, 1:1000 (WB) and 1:50 (IF)]; Novus Biologicals - rabbit anti-proteasome 20S alpha 5 antibody [cat # NBP1-86838, 1:500 (IF)], mouse anti-Proteasome 19S (PA700) antibody [cat # H00005701-M01, 1:50 (IF)]; Santa Cruz, mouse monoclonal anti-CHIP-STUB1 antibody [cat # SC-133083, 1:50 (IF)]; Cell Signaling - rabbit anti-PARP antibody [cat # 9542S, 1:1000 (WB)], rabbit anti-PA28β (PSME) and rabbit anti-PA28γ (PSME3) antibodies [cat # 2409 and # 2412, 1:25 (IF)], rabbit anti-K48-linkage Specific Polyubiquitin (D9D5) antibody [cat # 8081S, 1:200 (IF)]; AbCam - mouse monoclonal anti-Tau antibody (Tau-5) [cat # ab80579, 1:1000 (WB), 1:400 (IF)], recombinant rabbit anti-Dcp1a monoclonal antibody [cat # EPR13822, 1:500 (IF)], rabbit monoclonal anti-BAG2 antibody [cat# ab79406, 1:2000 (WB, Fig. 3g: endogenous); Provided by P. Davies, Feinstein Institute for Medical Research, Manhasset, NY – mouse anti-paired helical filament-1 (PHF-1) monoclonal antibody, and mouse anti-MC1 monoclonal antibody [1:1000 (WB)]; ThermoFisher - Phospho-Tau (Ser202, Thr205) monoclonal antibody (AT8) [cat # MN1020, 1:1000 (WB)], mouse polyclonal anti-PA28 γ (PSME3) antibody [cat# 89-006-918, 1:100 (IF)]; Sigma-Aldrich - mouse anti-β-Actin antibody [cat # A5441, 1:5000 (WB)]; BD Biosciences – mouse anti-p62 Ick ligand, [cat #610833, 1:200 (IF)], purified mouse anti-Human G3BP Clone 23/G3BP (RUO) [cat # 611126, 1:250 (IF)]; Enzo Life Sciences – mouse mono- and polyubiquitinylated

conjugates monoclonal antibody (UBCJ2) [cat # ENZ-ABS840-0100, 1:200 (IF)]; Invitrogen – Alexa fluor 488 goat anti rabbit (cat # A11008) or anti mouse (cat# A11001) IgG [1:450 (IF)], Alexa Fluor 555 goat anti-rabbit (cat # A21428) or anti-mouse (cat# A21422) IgG [1:450 (IF)], Alexa Fluor 790 goat anti-rabbit IgG [1: 5000 (WB)] and Alexa Fluor 680 goat anti-mouse IgG [1:5000 (WB)].

The following reagents were prepared in stock solutions 100X concentrated and added to the cell culture before imaging or other assays at the indicated final concentration: sucrose (Sigma-Aldrich, 125–500 mM), oligomycin (Sigma-Aldrich, 1 μM), MG132 (Sigma-Aldrich – 10 μM), lipopolysaccharide - LPS (Sigma-Aldrich, 10 μg ml$^{-1}$), sodium arsenite (NaAsO$_2$, Sigma-Aldrich, 500 μM), vinblastine (Sigma-Aldrich, 5 μM), MLN-7243 (Active Biochem, 1 μM), Bafilomycin A1 (Cell Signaling, 50nM). The SYTO™ 14 Green Fluorescent Nucleic Acid Stain (cat # S7576, ThermoFisher) was used according to the manufacturer instructions to stain nucleic acid in living cells.

The backbone plasmid Clover-mRuby2-FRET-10 (Addgene plasmid # 58169; http://n2t.net/addgene:58169; RRID:Addgene_58169) was used. It contains the P2A self-cleaving peptide between the mClover2 and mRuby2 tags. The human Tau full-length cDNA was fused to the C terminus of mRuby2 and the human BAG2-full length cDNA was fused to the C terminus of mClover2 tag to generate cells expressing both proteins. The same backbone plasmid was used to generate independent constructs: mRuby2-BAG2, mClover2-BAG2, mClover2BAG2-I160A, mClover2-BAG2-Δ20-61, mClover2-BAG2-S20E, mClover2-BAG2-S20A, and mRuby2-Tau. BAG2-full length cDNA was also cloned into pEYFP-C1 (Clontech, cat # 6006-1). Human TIA-1 cDNA was cloned into pEGFP-C1 (Clontech, cat # 632370). Human HSP70 cDNA were cloned into a pDsRed2-C1 (Clontech, cat # 632407). Ubiquitin-independent proteasome activity was assessed using the ZsProSensor-1 vector (Cat # 632425, Takara Bio Inc) named ZsGreen (+PEST). The same plasmid was modified to generate the ZsGreen (−PEST), used as a control.

**Cell culture, transfections, CRISPRi and stable cell lines**. SH-SY5Y neuro-blastoma cells were cultured in DMEM/F12. H4, H4i (constitutively expressing CRISPRi machinery with a TagBFP[21], HeLa and COS7 cells were grown in DMEM. All cells were supplemented with 10% FBS, 100 μg ml$^{-1}$ penicillin/streptomycin. Cultures were maintained in a humidified atmosphere of 5% CO$_2$ at 37 °C. Transfection were performed with Lipofectamine 3000 (Invitrogen) according to the manufacturer's instructions and cells were assayed after 24 h. Cells routinely tested negative for mycoplasma.

Single-guide RNA (used for dCas9-KRAB gene repression) constructs for individual genes were cloned into the pLG15 vector according to previously published methods[57] to generate lentivirus particles. H4i cells were then transduced with BAG2 sgRNA (5'-GTGGGGAGCGCAAGTCTCTG-3') or non-targeting sgRNA (NT or scramble shRNA) constructs and selected with puromycin (1 μg ml$^{-1}$). Knockdown was confirmed by quantitative western blot and ICC (Fig. 3g, h).

Stable SH-SY5Y cell line generation was made using the positive selection marker G418/Geneticin antibiotic resistance, which was delivered using the same plasmid containing the gene of interest (in *cis*). 24 h after cell transfection, the geneticin was added to the medium (500 μg ml$^{-1}$). Media was replaced every 2 days and cells were examined daily for visual toxicity. Typically, the cells that have undergone plasmid integration survive for 9 days post-transfection under antibiotic selection. Surviving cells were allowed to expand in a T75 flask for at least 4 weeks and then, they were used as a polyclonal line.

**BAG2 mutants' constructs**. The BAG2-Δ20-61, I160A, S20A and S20E mutants were generated by site-directed mutagenesis PCR using following primers: BAG2-Δ20-61 (forward: 5'-agtatccaaaatagccaggacatgagg-3'; reverse: 5'-ggaggagcggca-gaagcg-3'), I160A (forward: 5'-gctggctgtgctcttgaagatca-3'; reverse: 5'-tactatggatt-gaaacttctgatcaactgg-3'), S20A (forward: 5'-gctatggctgaccgctcca-3'; reverse: 5'-ggaggagcggcagaagcg-3' and S20E (forward: 5'-gaaatggctgaccgctcca-3'; reverse: 5'-ggaggagcggcagaagcg-3'). The intended mutations were verified by Sanger sequencing.

**FRAP analysis**. BAG2 transfected SH-SY5Y cells were culture in 8 well glass bottom μ-slides (Ibidi) with complete media. Fluorescence recovery after photo-bleaching (FRAP) experiments first identified a bleach area of ~2 μm. Then images were taken every 5 s during recovery. Mean intensity within similar sized areas was determined at each time point using Imaris software (Oxford Instruments). Mean intensity within a different part of the cytoplasm was also measured (IBt) to correct for bleaching during image acquisition. Corrected mean intensity at each time point was determined by taking the ratio: It/IBt. A curve for exponential recovery was determined using easyfrap (https://easyfrap.vmnet.upatras.gr/).

**Fluorescence correlation spectroscopy (FCS)**. All fluorescence correlation spectroscopy (FCS) measurements were performed on a Leica SP8 Resonant Scanning Confocal equipped with a cooled HyD detector. For each set of measurements, the excitation volume for a specific optical configuration was calibrated using Alexa 488 at 37 °C (D = 464.23 μm$^2$ s$^{-1}$). After fixing the diffusion coefficient for Alexa 488 calibrations, a structural parameter of 3.0, a lateral radius of 0.25 μm, and an axial radius of 0.74 μm were fit. Data were fitted to a normal 3D

diffusion model with triplet correction[23,24,58] as shown below:

$$G(\tau) = G(0)\left[1 - T + Te^{-\frac{\tau}{\tau_T}}\right]\left[1 + \left(\frac{\tau}{\tau_D}\right)^\alpha\right]^{-1}\left[1 + \frac{1}{\kappa^2}\left(\frac{\tau}{\tau_D}\right)^\alpha\right]^{-\frac{1}{2}} + G_\infty \quad (3)$$

$$\kappa = \frac{z_0}{\omega_0}, \quad (4)$$

$$V_{Eff} = \pi^{3/2}\omega_0{}^2 z_0, \quad (5)$$

$$\langle C \rangle = \frac{\langle N \rangle}{V_{Eff} N_A}, \quad (6)$$

$$D = \frac{\omega_0{}^2}{4\tau_i}, \quad (7)$$

where $\tau_D$ is the diffusion time, α is the anomaly parameter, $\tau_T$ is the lifetime of the triplet state, $T$ is the triplet amplitude, $\kappa$ is the structural parameter for the focal volume, $G_\infty$ is the correlation offset, $N$ is the average number of molecules in the focal volume, $V_{Eff}$ is the effective excitation volume (estimated by the calibration measurement), $N_A$ is Avogadro's number, $C$ is the concentration of molecules in the focal volume, $D$ is the diffusion coefficient, $\omega_0$ is the effective lateral focal radius at e^−2 of its maximum value, and $z_0$ is the effective focal radius along the optical axis at e^−2 of its maximum value.

The concentration to intensity calibration curve was constructed by transiently transfecting mClover2 BAG2 at low expression levels in SH-SY5Y cells and measuring the FCS at multiple points. The resulting data was fit to a linear model and used for calculating the concentrations of mClover2-BAG2 fusion protein (Supplementary Fig. 5a). An example of the raw data and fits is provided (Supplementary Fig. 5b). All raw calibration data are included in the "Source Data file: Supplementary Information". It is important to note that several factors can have large effects on the concentration estimates, leading to an accuracy with 2-fold of the actual values[59]. These effects include photobleaching, protein maturation times, and other optical artifacts including cover-slip thickness variation, optical saturation, and other aberrations[60,61].

Analysis of BAG2 concentrations and transfer free energies were adapted from studies by the Brangwynne lab[25]. For determining the average [BAG2]$_{Dense}$ droplet concentrations, individual cells were masked in Fiji (Version 1.53c) and exported to ilastik (Version 1.3.3post2)[62] for particle and object selection. After training on a representative set of cells, the entire dataset was processed, and individual droplet identities were exported and analyzed in MATLAB version R2019B. The average concentration of each droplet was determined and from this the average droplet concentration per cell was calculated and used for subsequent analyses.

For determining the [BAG2]$_{Dilute}$ phase concentrations, both a nucleus and droplet mask were applied to the cells to leave just cytoplasmic intensity. A moving average in the size of a 25 × 25 pixel box was applied to the image and the 25% quartile was determined and used as the dilute phase concentration. [BAG2]$_{Total}$ was calculated similarly by only masking the cell nucleus and calculating the average across the cell.

The transfer free energy was calculated by using the Eq. (2), where (1) is the partition coefficient for BAG2 in and out of condensates. A pairwise $t$-test was used to compare ΔGtr values before and after sucrose treatment. Plots of [BAG2]$_{Dilute}$ versus ΔGtr were fit to the model (8),

$$\triangle G^{tr} = \frac{(k \times C^{max})}{(1 + k \times [BAG2]^{Dilute})}, \quad (8)$$

using fitnlm in MATLAB version R2019B and plotted with 90% mean confidence intervals.

**FLIM-FRET analysis**. SH-SY5Y cells expressing the fluorescent pair proteins (mClover2 and mRuby2) were plated in 8 well glass bottom μ-slides (Ibidi) and exposed to complete media. The mClover2-mRuby2 FRET pair was chosen for the long lifetime of the donor (clover) and suitable FRET efficiency between the pair[35]. The full construct linked the FRET pair with P2A that self-cleaves[34]. The single cell analysis was performed using Fluorescence Lifetime Imaging Measurements (FLIM) on a Leica SP8 microscope equipped with a HyD detector with a 63× (1.49NA, oil) objective. A white-light laser emitting at 488 nm was used for excitation and the emission was monitored from 508 to 545 nm. The images were acquired with 16 sequential line scans with a scan speed of 100 ms. FRET was analyzed with the FLIMfit software tool developed at Imperial College London (Version 5.1.1)[63]. A control cell line expressing a mClover2-BAG2-P2A-mRuby2 vector did not display a bimodal distribution of fluorescence lifetimes and fit well to a single exponential decay with a lifetime of 3.2 ns indicating that there was no association between clover-BAG2 and free Ruby and was in good agreement with reported values[64]. Subsequently, experimental data was fit to double exponential decays, the high lifetime component was either fixed at 3.2 ns or left to be a free parameter. In either case, the fits converged to similar values. When BAG2 and Tau come within less than 63 Å, FRET will be observed as indicated by a drop in the measured fluorescent lifetime. Because any given pixel of an image is composed of both BAG2$_{Free}$ and BAG2$_{Bound:Tau}$, we fitted the associated pixelwise decays to a

double exponential, with the higher lifetime representing the BAG2$_{Free}$ fraction, and the lower exponential representing the BAG2$_{Bound:Tau}$. By observing the weighted mean lifetime at each pixel, the relative amount of FRET occurring was determined. The lowest observed lifetimes were $2.8 \pm 0.1$ ns, a value that matches the value of a fused clover-mRuby2 construct (Martin et al.[35]). This value indicated the case with the highest efficiency FRET observed in our system. Values between 2.8 and 3.2 ns represent cases in which only a fraction of the total BAG2 was bound, or alternatively times when the distance between BAG2 and Tau was farther apart and therefore had lower FRET efficiency.

**Immunofluorescence**. After treatments, cell cultures were fixed in a 1:1 methanol-acetone for 10 min at $-20\,°C$, washed ($3\times1$ min) with PBS, and permeabilized with 0.1% Triton X-100 for 15 min. Cells were washed $2\times$ in PBS and incubated for 30 min in blocking buffer at RT (2% NGS, normal goat serum, 4% BSA, 0.2% Triton X-100). Cells were incubated with primary antibody in blocking medium overnight at $4\,°C$, washed ($3\times5$ min) with PBS and incubated for an additional 1 h with secondary antibodies. After washing ($3\times5$ min), the dishes were covered in antifade mounting media (VECTASHIELD® Antifade Mounting Media, Vector Laboratories) and visualized by a Leica SP8 fluorescence microscopy.

**Cellular viability measurements**. Cellular viability was determined after 2 h of sucrose exposure in cells transiently expressing BAG2 full length or BAG2 mutants (BAG2Δ20-61 or I160A). Cellular viability was measured using Cell titer-Glo Luminescent cell Viability Assay kit (Promega) according to the manufacture's instruction. SH-SY5Y cells were plated in opaque-walled 96 well plate and exposed to complete media containing 500 mM of sucrose for 2 h. Following exposure, the cell titer-Glo luminescent reagent was added and incubated for 10 min at RT. Luminescence measurements were performed using a Victor3 plate reader (PerkinElmer; Waltham, MA).

**Protein extraction and western blot**. Western blotting assay was performed after sucrose treatment. Briefly, after treatments, cells were lysed with RIPA buffer (1% Triton X-100, 0.5% sodium deoxycholate, 0.1% SDS, 150 mM NaCl, 50 mm TrisHCl, pH 7.4). Protein concentration was estimated by the BCA protein assay kit (Pierce) and was adjusted to $1\,\mu g\,\mu l^{-1}$. Total cell lysates (10 $\mu g$) were resolved by SDS-PAGE (Mini-Protean TGX Precast Gels, BioRad, cat# 456-9036) and transferred to a 0.45 $\mu m$ nitrocellulose membrane (GE Healthcare, cat # 10600007). Membranes were incubated in a blocking buffer (PBS, 5% BSA) for 1 h at RT. After overnight incubation ($4\,°C$) with primary antibody, the blots were washed $2\times$ in Tween 20-TBS (TBS-T) then incubated with IRDye secondary antibodies (1:5000 dilution in LI-COR blocking buffer) for 1 h at RT. The blots were then washed $2\times$ in TBS-T and scanned using a digital fluorescent imaging system (Odyssey CLX, LI-COR). Quantification of pixel intensity was done with ImageJ and bands were quantified as a percentage of the control. Beta-actin was used as a housekeeping control.

**Recombinant protein purification**. Proteins used for binding and nucleotide release were purified as described previously[65]. Briefly, BAG2 and BAG2 mutant expressing bacteria were pelleted, re-suspended in His Binding Buffer (50 mM Tris, 300 mM NaCl, 10 mM Imidazole pH 8.0) + protease inhibitor tablets (Roche), and then sonicated. Supernatants were incubated with Ni-NTA resin for 2 h at $4\,°C$, washed with Binding Buffer, His Washing Buffer (50 mM Tris, 300 mM NaCl, 30 mM Imidazole pH 8.0) and finally eluted with His Elution Buffer (50 mM Tris, 300 mM NaCl, 300 mM Imidazole pH 8.0 After Ni-NTA columns, all proteins were subjected to TEV cleavage overnight and dialyzed into MonoQ Buffer A (20 mM HEPES, 10 mM NaCl, 15 mM β-ME, pH 7.6). Proteins were applied to a MonoQ column (GE Healthcare) and eluted by a linear gradient of MonoQ Buffer B (Buffer A + 1M NaCl). Fractions were concentrated and applied to a Superdex S200 (GE Healthcare) size exclusion column in BAG buffer (25 mM HEPES, 5 mM MgCl$_2$, 150 mM KCl pH 7.5). Hsp72 was purified as described elsewhere[66]. To make apo-Hsp72, the protein underwent extensive dialysis; day one (25 mM HEPES, 100 mM NaCl, 5 mM EDTA pH 7.5), day two (25 mM HEPES, 100 mM NaCl, 1 mM EDTA pH 7.5), day three (25 mM HEPES, 5 mM MgCl$_2$, 10 mM KCl pH 7.5). BAG2 proteins were labeled with Alexa Fluor® 488 5-SDP ester (Life Technologies) according to the suppliers' instructions. Hsp72 was biotinylated using EZ-link NHS-Biotin (Thermo Scientific) according to the supplier instructions. After labeling, the proteins were subjected to gel filtration to remove any unreacted label. Average label incorporation was between 1 and 2.0 moles of label per mole of protein, as determined by measuring fluorescence and protein concentration (A$_{max}$ × MW of protein/[protein] × ε$_{dye}$).

**Flow cytometry protein interaction assay (FCPIA)**. The assay procedure was run according to previously published protocols[65]. Briefly, biotinylated Apo-Hsp72 was immobilized (1 h at room temperature) on streptavidin coated polystyrene beads (Spherotech). After immobilization, beads were washed to remove any unbound protein and then incubated with labeled BAG2 protein at indicated concentrations. Binding was detected using an Accuri™ C6 flow cytometer to measure median bead-associated fluorescence. Beads capped with biocytin were used as a negative control, and non-specific binding to beads was subtracted from signal.

**Nucleotide release assay**. Nucleotide release from Hsp70 was performed as described previously[46,65]. Briefly, a fluorescent ATP analog, N6-(6-Amino)hexyl-ATP-5-FAM (ATP-FAM) (Jena Bioscience) was used to measure BAG2 induced nucleotide dissociation from Hsp72. In black, round-bottom, low-volume 384-well plates (Corning), 1 μM Hsp72 and 20 nM ATP-FAM were incubated with varying concentrations of BAG2 protein for 10 min at room temperature in assay buffer (100 mM Tris, 20 mM KCl, 6 mM MgCl$_2$ pH 7.4). After incubation fluorescence polarization was measured (excitation: 485 nm emission: 535 nm) using a SpectraMax M5 plate reader.

**Imaging and data analysis**. Imaging data (both in vivo and ex vivo) were acquired using Resonant Scanning Confocal microscopy (Leica SP8) with the corresponding laser wavelengths sequentially scanned by line. For live imaging an environmental chamber with 5% of CO$_2$ and 37 °C temperature was used. After performing experiments, images were visualized and analyzing in Leica application Suit X and Imaris software (v. 9.7.2; Oxford Instruments), unless otherwise noted. At least five condensates were required to designate a cell as containing a dense phase. Fluorescence quantification, granule area sum, number and size of condensates, and colocalization were quantified using Imaris. Size detection lower than 0.20 μm (about the maximum confocal resolution) were cut off from the analysis. Image-deconvolution algorithms were applied to remove the out-of-focus blur typical for epifluorescence images and improve both lateral and axial resolution. Images were deconvolved with Huygens Essential for Win64 version 18.10 (Scientific Volume Imaging, the Netherlands, http:svi.nl), using the CMLE algorithm, with SNR:20 and 40 iterations.

**Statistical analysis**. All statistical analyses were performed using Prism 6 software. Data were analyzed for statistical significance using the following methods: one-way ANOVA followed by Dunnett's or Tukey`s multiple comparison post-test; two-way ANOVA followed by Tukey`s multiple comparison post-test, or Student's $t$ test. $p$ values < 0.05 were considered statistically significant.

**Reporting summary**. Further information on research design is available in the Nature Research Reporting Summary linked to this article.

## Data availability
The data supporting the findings of this study are contained within and provided in the Source Data file.

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

## Acknowledgements

This study was funded by U54 NS (100717; K.S.K.), the Dr. Miriam and Sheldon G. Adelson Medical Research Foundation (K.S.K.), the Larry L. Hillblom Foundation (K.S.K.), W.M. Keck Foundation (K.S.K.) and the National Institutes of Health (R01AG056058; K.S.K. and NS059690; J.E.G.). D.C. Carrettiero and M.C. Almeida acknowledge the São Paulo Research Foundation, FAPESP (2019-22708-1 and 2019-22819-8). We acknowledge the use of the Neuroscience Research Institute, Molecular, Cell and Developmental Biology Microscopy Facility, and the Resonant Scanning Confocal Microscope supported by the National Science Foundation Major Research Instrumentation Program (DBI-1625770).

## Author contributions

Author contributions: K.S.K. and D.C.C. conceived the project. D.C.C., M.C.A., A.P.L., D.H. and X.Z. performed the cellular experiments. K.S.K. supervised the work. J.N.R. performed and J.E.G. supervised the in vitro experiments. S.N. advised on protein structures and FCS computations. K.S.K. and D.C.C. wrote the manuscript. All authors edited and reviewed the manuscript.

## Competing interests

The authors declare no competing interests.
