## [Peer Review File · Nature Communications]

Stress Routes Clients to the Proteasome via a BAG2 Ubiquitin-Independent Degradation CondensateREVIEWER COMMENTS

Reviewer #1 (Remarks to the Author):

The present manuscript elucidates the properties and function of a membrane-less compartment formed by the co-chaperone BAG2. It is shown that BAG2 forms condensates that are distinct from other membrane-less compartments such as stress granules and p-bodies. Condensate formation is mainly triggered by hyperosmotic stress but also by heat stress and proteasome inhibition. Analysis of BAG2 mutant variants reveals the dependence of condensate formation on BAG2 oligomerization and interaction with Hsp70. Condensates seem to recruit 20S proteasomes and the proteasome activator PA28 to mediate the ubiquitin-independent degradation of client proteins.

The findings extend previous work by the Kosik lab (Carrettiero et al., 2009), which already described BAG2 condensates as compartments for the ubiquitin-independent degradation of chaperone clients (i.e. tau) and also elucidated some of their biophysical properties. I am not sure whether the described new data represent a sufficient conceptual advancement to justify publication in Nature Communications. Concerns also relate to the lack of degradation assays in BAG2 depleted cells, which would reveal the extent and relevance of BAG2-mediated ubiquitin-independent protein degradation under stress. Finally, molecular details of proteasome and PA28 recruitment as well as client selection remain elusive.

1. Extended Data Fig. 1a: The authors show that BAG2 condensates are increased in stably transfected cells under various stress conditions. As overexpression of BAG2 contributes to condensate formation, it remains unclear whether endogenous BAG2 would show a similar response to the applied stress conditions. The authors should monitor condensate formation of endogenous BAG2 in response to the different stress conditions, expanding the data shown in Fig. 1h and i for sucrose treatment.
2. Fig. 3e and f: The authors deduce from the shown data that BAG2 condensates protect cells against stress. Yet, all data rely on the overexpression of BAG2. Depletion of endogenous BAG2 needs to be performed to verify an essential role of BAG2 in stress protection.
3. Fig. 4e: The authors claim to detect 20S proteasomes here, but the used antibody against a 20S proteasome subunit would also detect 26S proteasomes. Labelling is apparently misleading and should be replaced by mentioning the detection of the subunit alpha 5.
4. Fig. 4e: To verify that BAG2 specifically recruits 20S proteasomes into condensates, the authors should use an antibody against 19S cap subunits and demonstrate the exclusion of 19S caps.
5. Fig. 4e and f: Colocalization of proteasomes and PA28 with endogenous BAG2 needs to be shown.
6. The molecular basis for PA28 and 20S proteasome recruitment into BAG2 condensates remains unclear. Is there a direct interaction between BAG2 and PA28 or the 20S core particle?
7. Extended Fig. 4d: There is no control demonstrating that the E1 inhibitor is working at all.
8. Fig. 5: The authors show that BAG2 condensates induced by overexpression of BAG2 cause a reduction of the levels of a ubiquitin-independent degradation reporter. It remains unclear, however, whether BAG2 at endogenous levels promotes the proteasomal degradation of such a reporter. BAG2 depletion experiments need to be performed.
9. The title states that 'Stress Routes Clients to the Proteasome via a BAG2 Ubiquitin-Independent Degradation Condensate'. Yet, there are no data showing that depletion of endogenous BAG2 stabilizes chaperone clients under different stress conditions. The extent and relevance of BAG2-mediated ubiquitin-independent degradation under stress remains elusive.

10. Tau is apparently degraded by a BAG2-mediated pathway but also by a CHIP-mediated one that is inhibited by BAG2. It remains unclear whether CHIP clients are generally targeted onto the BAG2 pathway upon BAG2 overexpression or whether BAG2 selects specific client proteins.

11. Fig. 6 mainly expands on previous findings of the Kosik lab (Carrettiero et al. 2009), where interaction of BAG2 with Tau in BAG2 condensates along microtubules has already been demonstrated.

12. Fig. 7c: The authors state that bafilomycin has no effect on BAG2 condensates. However, in the presented picture BAG2 looked much more diffusely distributed and seemed to display reduced condensate formation upon bafilomycin treatment. Although BAG2-mediated degradation doesn't seem to involve autophagy, there still seems to be a crosstalk between these pathways.

13. Fig. 7f: The picture doesn't show p62 accumulation in regions devoid of BAG2 as stated by the authors. It rather seems that p62 is concentrated in a portion of the BAG2 condensate. Segregation of BAG2 and p62 cannot be deduced from the presented data.

Reviewer #2 (Remarks to the Author):

Carrettiero et al. investigate the formation of membraneless organelles by BAG2. Using imaging and spectroscopy they demonstrate that BAG2 containing organelles are formed under stress, that they are ubiquitin independent, that the organelles align with microtubules and that they degrade tau protein. Using Fluorescence Recovery after Photobleaching (FRAP) they demonstrate that the organelles exchange BAG2 with the cytosol. They calibrate their concentration-intensity relation by Fluorescence Correlation Spectroscopy (FCS) and then use image analysis to derive monomeric BAG2 and BAG2 condensate concentrations to determine the transfer free energy of solution and condensates. Their analysis before and after stress indicates that heterotypic interactions increase after stress in BAG2 condensates, although it is not clear what drives these changes. The manuscript does not contain sufficient information to judge the use of FCS. The authors should include:

- FCS calibration and measurement curves including fits, fit parameters and residuals.
- the concentration intensity calibration derived from FCS

Where FCS calibrations and measurements performed with the same fit model? Did the cell measurements show only one diffusive component?

Minor issues:

Line 309: Please rewrite the sentence: "... BAG2 phase separation in cells with by showing ..."

Line 318 and 383: replace "Zsgreen" with "ZsGreen" for consistency.

Line 421: replace " $\mu\text{M}^2/\text{sec}$ " with " $\mu\text{m}^2/\text{s}$ "

Reviewer #3 (Remarks to the Author):

The work of Carrettiero et al focuses on a BAG2 Ubiquitin-Independent Degradation condensate. The authors report that BAG2 is a marker for a membraneless organelles triggered by stress but distinct from stress granules and processing bodies. BAG2 appears to lack RNA and ubiquitin, while favors

degradation of client proteins via the 20S proteasome. In particular, this new type of membraneless organelles respond specifically to hyperosmotic stress.

Overall the work is of significance for the field and the results appear new in the field, since previous work reported BAG2 only as a component of phase-separated organelles, but not as one of the main component associated to hyperosmotic stress granules (see Esposito et al., 2021)

The work supports the main findings that BAG2 condensation respond to hyperosmotic stress favoring phase separation, whereas arsenite-induced stress does reduce the condensation of the protein. Main markers of stress granules and processing bodies are not found in colocalization with BAG2, supporting the protein is part of a distinct type of organelle. Finally, the condensates seems to be lacking RNA content. In terms of condensation, BAG2 phase separation requires both the coiled-coil and BAG2 domain.

There are few concerns that may need to be addressed:

- BAG2 Condensates increase in size with hyperosmotic stress. However, hyperosmotic stress causes an increase in the cell concentration by altering the volume of the cell. If that is the case, the change in size (but not in number) of the condensates would represent just a shift in the total concentration of the components. It would be helpful to provide an estimate of the variation in the cell size before and after hyperosmotic stress. If a difference in volume is observed, it can be used to estimate the phase diagram of the component as a function of the total protein concentration.

- FCS is used to determine the concentration inside and outside the dense phase of the organelles but no example of correlation curves, FCS calibration and fit is reported besides the ratio of the final concentrations. This makes impossible to judge the quality of the source data. It would be also important to report the observed changes in the diffusion coefficient and number brightness since they report about the local viscosity perceived in the cell, possible aggregation or quenching effects. Finally, the reference to the method is actually making use of specific TAG-lens that the authors do not mention and that possibly is not applied to their experiments. Adding or substituting the citation for other works that harness FCS to study biomolecular condensates may provide a more reliable sources.

- When plotting the dependence of the ΔG on the dilute concentration, one would expect the values to tend to zero at low and high protein concentration, since the limiting conditions coincide with the transition to a single phase. One would expect this to be true even if the phase separation is heterotypic, unless BAG2 is recruited to the condensate. The authors may want to expand on their reasoning on how that type of curvature can be obtained and what it implies in terms of the condensate composition and role of BAG 2. E.g. is BAG2 a client or a scaffold in these condensates?

- Have the authors considered changing temperature to explore the phase diagram of the protein? It seems that FCS provides boundaries for both the dense and light phase. Temperature, which sets the strength of interactions, should provide a mean to see how these concentrations changes and to identify whether increasing or decreasing temperature is favoring the phase separation.

- FRET constructs for BAG2 and tau interaction detail a peculiar territorial organization of the components in the dense phase. Is this stable over time or does it change?

- Are the BAG2 condensates specific to the studied cell line? or do they appear in other cell lines too?

- Is the response of BAG2 condensate to hyperosmotic stress specific to sucrose or other osmolytes can cause the same effect? For example, addition of small PEG molecules (e.g PEG 400) has been shown to cause collapse of cells - does this provide a similar result than sucrose?

- When mentioning that oligomerization would lower the barrier for LLPS (line 126) the authors may want to explicit their reasoning (e.g. increased multivalency).

NCOMMS-19-11141432

In response to the editorial and the reviewers' comments, we have made substantial revisions to our manuscript. We thank for the in-depth review of our manuscript and the constructive criticism helping us to improve the quality of our work.

Below you can find a point-by-point responses raised:

Reviewers' comments:

Reviewer #1:

-The present manuscript elucidates the properties and function of a membrane-less compartment formed by the co-chaperone BAG2. It is shown that BAG2 forms condensates that are distinct from other membrane-less compartments such as stress granules and P-bodies. Condensate formation is

mainly triggered by hyperosmotic stress but also by heat stress and proteasome inhibition. Analysis of BAG2 mutant variants reveals the dependence of condensate formation on BAG2 oligomerization and interaction with Hsp70. Condensates seem to recruit 20S proteasomes and the proteasome activator PA28 to mediate the ubiquitin-independent degradation of client proteins.

The findings extend previous work by the Kosik lab (Carrettiero et al., 2009), which already described BAG2 condensates as compartments for the ubiquitin-independent degradation of chaperone clients (i.e. tau) and also elucidated some of their biophysical properties. I am not sure whether the described new data represent a sufficient conceptual advancement to justify publication in Nature Communications.

Authors: *We thank the reviewer for noting Carrettiero et al., 2009. The current paper builds on that original data and extends the work in unexpected and novel directions. We are describing a novel cellular entity, the BAG2 Ubiquitin-Independent Degradation Condensate. This liquid-liquid phase separated membraneless organelle condenses in the cytoplasm in response to stress and includes a set of components that collaborate to implement a protein refolding/degradation decision. The description of this new cellular entity and its function fills a missing aspect of how the cell handles protein degradation in the cytoplasm under stress conditions. The conceptual advances presented will likely be of broad interest to the cell biology community. Our original paper (Carrettiero et al., 2009) does not report BAG2 condensates; whereas the current paper provides extensive evidence – both in vivo and in vitro- for a novel condensate labeled by BAG2, as shown by several immunocytochemical and biophysical techniques. Furthermore, we have utilized advanced cell biological techniques such as FLIM-FRET and fluorescent correlation microscopy (FCS) to probe the components of this complex multi-component structure. The distinction from other well-known condensates, such as stress granules and P bodies, that like the BAG2 condensates, can be considered membraneless organelles, is presented in detail. Specifically, the relative absence of RNA and ubiquitin in these structures distinguishes them from those related membraneless organelles. Interestingly, we show the relationship of BAG2 condensates to stress in a manner that further distinguishes these organelles from stress granules as they are not induced by arsenite. Functionally, BAG2 condensates are capable of protecting the cell from stress and as elaborated upon below, mediate ubiquitin-independent degradation.*

-Concerns also relate to the lack of degradation assays in BAG2 depleted cells, which would reveal the extent and relevance of BAG2-mediated ubiquitin-independent protein degradation under stress. Finally, molecular details of proteasome and PA28 recruitment as well as client selection remain elusive.

Authors: *These points are addressed by new experiments detailed below. We thank the reviewer for the in-depth review of our manuscript and the constructive criticism that has improved the quality of our work.*

1. Extended Data Fig. 1a: The authors show that BAG2 condensates are increased in stably transfected cells under various stress conditions. As overexpression of BAG2 contributes to condensate formation, it remains unclear whether endogenous BAG2 would show a similar response to the applied stress conditions. The authors should monitor condensate formation of endogenous BAG2 in response to the different stress conditions, expanding the data shown in Fig. 1h and i for sucrose treatment.

Authors: *We monitored condensate formation of endogenous BAG2 in response to hyperosmotic stress, as well as other factors that induce stress or related conditions such as temperature, LPS, MG132, and oligomycin. These results are presented as follows: Other stressors such as temperature (42°C), LPS, oligomycin and MG132 also increase endogenous BAG2 condensation (**Extended Data Fig. 2 and main Fig 7d. Also, in result section – lines 103-104 and line 311-315**)*

2. Fig. 3e and f: The authors deduce from the shown data that BAG2 condensates protect cells against stress. Yet, all data rely on the overexpression of BAG2. Depletion of endogenous BAG2 needs to be performed to verify an essential role of BAG2 in stress protection.

Authors: *Thank-you for suggesting this important addition to the manuscript. Single-guide RNA (sgRNA, used for dCas9-KRAB gene repression) constructs for the BAG2 gene were cloned into the pLG15 vector and lentiviral particles were generated. H4i cells, constitutively expressing CRISPRi machinery, were then transduced with BAG2 sgRNA or scramble lentiviral construct and selected with puromycin. BAG2 knockdown was confirmed by western blotting in the “**main figure 3g - blots**” and by ICC in the “**main figure 3h - images**”. The presence of BAG2 condensates and its response to sucrose was also confirmed in H4 cell line and it is shown in the “**main figure 3i - images**”. The role of BAG2 in stress protection was then verified by western blot analysis using anti-PARP antibody (cleaved and non-cleaved species). Data are shown in the “**main figure 3g - plot**”. These results demonstrate that BAG2 depleted cells are more*

*sensitive to stress than control cells (NT sgRNA). The results are presented in the main text “**result section – lines 170-177**” and in the “**Method section – lines 425 – 437**”*

3. Fig. 4e: The authors claim to detect 20S proteasomes here, but the used antibody against a 20S proteasome subunit would also detect 26S proteasomes. Labelling is apparently misleading and should be replaced by mentioning the detection of the subunit alpha 5.

Authors: *We replaced the labeling in the “**main figure 4e**” to “subunit alpha 5” instead of “20S proteasome”. The description was changed in the main text “**result section – lines 213**” and respective figure legends. We also included a brief description concerning this point in “**result section – lines 219-220**”. Additional experiments as suggested in comment #4 verified that BAG2 specifically recruits the 20S proteasome into condensates.*

4. Fig. 4e: To verify that BAG2 specifically recruits 20S proteasomes into condensates, the authors should use an antibody against 19S cap subunits and demonstrate the exclusion of 19S caps.

Authors: *We thank the referee for this excellent suggestion. To verify that BAG2 specifically recruits the 20S and not the 26S proteasome into condensates we performed an ICC for endogenous BAG2 and endogenous 19S caps also called PA700 (anti-Proteasome 19S S7, H00005701-M01, Novusbio – added in the “**Method section – lines 383-384**”). The exclusion of 19S caps from BAG2 condensates was very pronounced. These data are shown in the “**extended figure 6c**”. A description was added in the main text “**result section – lines 220-222**”.*

5. Fig. 4e and f: Colocalization of proteasomes and PA28 with endogenous BAG2 needs to be shown.

Authors: *As suggested, to verify that BAG2 specifically recruits the PA28 gamma regulatory cap into endogenous condensates we performed ICC for both endogenous BAG2 and PA28 gamma (antibody against PA28 gamma, 89-006-918 Fisher Scientific – added in the “**Method section – lines 394,395**”). Colocalization between BAG2 condensates and PA28 cap was clearly demonstrated. Data are shown in the “**extended figure 6d**”. In accord with the reviewer suggestion, we also included a brief description for the reader regarding the importance of verifying the results in an endogenous system. Description was added in the main text “**result section –lines 222-224**”.*

6. The molecular basis for PA28 and 20S proteasome recruitment into BAG2 condensates remains unclear. Is there a direct interaction between BAG2 and PA28 or the 20S core particle?

Authors: *This is a very interesting point. We have no evidence of a direct molecular interaction between BAG2 and PA28 gamma or 20S core particle. As a liquid–liquid phase separated entity it is more likely that multivalent macromolecular interactions drive the transition to a condensed phase. In our case, condensation involves several distinct partners of the degradation/refolding system such as BAG2, Hsp70, 20S proteasome, PA28 gamma and the client Tau in a dynamic way. The nature of these interactions in this crowded environment are of interest for additional biophysical studies. “Introduction section – lines 68 - 70” and a more in-depth description “Result section – lines 128-132”*

7. Extended Fig. 4d: There is no control demonstrating that the E1 inhibitor is working at all.

Authors: *The reviewer is correct, the control was missing. To verify if the effectiveness of the E1 inhibitor, SH-SY5Y cells were pre-treated with the ubiquitin E1 inhibitor MLN-7243 (1 μ M, 1h) and subjected to hyperosmotic stress and then verified by western blot analysis using anti-K48-linkage Specific Polyubiquitin antibody. The pre-treatment with the E1 inhibitor prevented the generation of different high polyubiquitinated bands after sucrose treatment. Data are shown in the “extended data Figure 7c”. Description was added in the main text “result section – lines 241 - 245”.*

8. Fig. 5: The authors show that BAG2 condensates induced by overexpression of BAG2 cause a reduction of the levels of a ubiquitin-independent degradation reporter. It remains unclear, however, whether BAG2 at endogenous levels promotes the proteasomal degradation of such a reporter. BAG2 depletion experiments need to be performed.

Authors: *As described above, we have performed several additional studies of endogenous BAG2 and BAG2 depletion experiments. However, for this particular point there is a challenge in using the reporter to clarify whether BAG2 condensates at endogenous levels promote proteasomal degradation. The depletion approach allows us to verify BAG2 function, but not the function of BAG2 condensation. Because BAG2 is present in a two-phase regime (a condensed dense phase and a dilute phase), we cannot deplete only the dense phase to check for the ubiquitin-independent degradation within the condensates.*

9. The title states that 'Stress Routes Clients to the Proteasome via a BAG2 Ubiquitin-Independent Degradation Condensate'. Yet, there are no data showing that depletion of endogenous BAG2 stabilizes chaperone clients under different stress conditions. The extent and relevance of BAG2-mediated ubiquitin-independent degradation under stress remains elusive.

Authors: *The stabilization of chaperone clients likely depends not just on BAG2, but in the properties of the condensate itself. As pointed out in #6, as a liquid–liquid phase separated entity it is presumed that multivalent macromolecular interactions drive the transition to a condensed phase under different stress conditions. In our experiments, condensation involved several distinct partners of the degradation/refolding system including BAG2, Hsp70, 20S proteasome, PA28 gamma and the client Tau all in a dynamic way. The nature of these multiple interactions in a crowded environment will require far more extensive biophysical experiments. However, as described in #8, the depletion approach allowed us to verify BAG2 function, but extrapolating that data to a condensate is an indirect conclusion.*

10. Tau is apparently degraded by a BAG2-mediated pathway but also by a CHIP-mediated one that is inhibited by BAG2. It remains unclear whether CHIP clients are generally targeted onto the BAG2 pathway upon BAG2 overexpression or whether BAG2 selects specific client proteins.

Authors: *This is an interesting and important question. We performed ICC for endogenous BAG2 and endogenous CHIP to clarify the complex partnership of the CHIP-mediated pathway in contrast to BAG2-mediated degradation. Data are shown in the “**extended data Figure 8**”. Only a relatively minor fraction of the BAG2 condensates colocalize with CHIP. In a small fraction (5%) of the largest BAG2 condensates (mean size: $0.75 \pm 0.08 \mu\text{m}$) selected from the total pool, ~80% of BAG2 condensates colocalized with CHIP. We speculate that small BAG2 condensates can mediate degradation/refolding pathways independently of CHIP but when the condensate gets larger, in some cases CHIP can be recruited. A description of these data was added in the main text “**result section – lines 235, 238**”. We have added some discussion about this observation “**result section – lines 238,239**”.*

11. Fig. 6 mainly expands on previous findings of the Kosik lab (Carrettiero et al. 2009), where interaction of BAG2 with Tau in BAG2 condensates along microtubules has already been demonstrated.

Authors: *The figure is the basis for our insights concerning dynamic features of BAG2 and Tau. The association of Tau with microtubules has long been viewed as a two-state dynamic in which Tau is either on or off the microtubule. The co-localization of Tau with BAG2 condensates along microtubules suggests a previously unrecognized off state: an exchange between microtubule-bound tau and BAG2 condensates. When off the microtubule but still in proximity to the microtubule, Tau can undergo LLPS in association with BAG2 condensates “**result section – lines 259, 260, 270-273**”. A related observation is that Tau preferentially associates with BAG2 condensates relative to the dilute phase when comparing cell populations with large numbers of BAG2 condensates to cells in same culture in which BAG2 is predominantly in the dilute phase (**Fig 6a, arrow vs arrowhead**). Cells with BAG2 condensates had a 1.8 ± 0.6 -fold increase in Tau levels compared to cells in which BAG2 was only present in the dilute phase ($p < 0.0001$; **Fig 6a, graph**). Furthermore, when microtubules were depolymerized with vinblastine, thereby preventing Tau from associating with microtubules, BAG2 condensates increased (**extended data Figure 9d**) possibly as a means to maintain the phase separation of a larger pool of Tau free of microtubules “**result section – lines 274-277**”. Interestingly, BAG2 condensates can find Tau (or traffic to Tau) and normally do so when Tau is in an on-off dynamic exchange with microtubules. Although outside the scope of this paper, we consider BAG2 condensates a mechanism to safely harbor Tau from fibrillization and create a reserve pool of Tau in proximity to the microtubule available for rapid changes in microtubule stability rather than diffusing further off into the cytoplasm.*

*Under stress conditions the cell directs BAG2 condensates to Tau. Endogenous BAG2 and Tau also increased in their association following hyperosmotic stress (**Extended Data, Fig 10a**). Confirmation of these endogenous observations utilized SH-SY5Y cells stably expressing clover-BAG2 and mRuby-Tau joined by the self-cleaving peptide P2A so that Tau and BAG2 localized independently after their translation. In the absence of stress, BAG2 was mainly observed in the dilute phase with few condensates (**Fig 6c, left images**). Hyperosmotic stress (15 min) rapidly induced BAG2 condensates that aligned along microtubules (**Fig 6c, right images**). These experiments demonstrate that stress can direct BAG2 condensates to Tau.*

*The BAG2/Tau interaction was further studied by FLIM-FRET measurements using the clover-mRuby2 FRET pair (**Fig. 6d**). After hyperosmotic stress and segmenting the images, the lifetimes of those pixels that belonged exclusively to BAG2 condensates was significantly lower than whole control cells ($p < 0.0001$) or whole cells after stress ($p = 0.027$) (**Fig 6d**). These data suggest that the majority of BAG2-Tau interactions occur within condensates.*

Finally, we sought direct detection of increased Tau degradation by BAG2 under hyperosmotic stress. SH-SY5Y cells expressing either Tau or Tau and BAG2 were subjected to 2h of hyperosmotic stress and probed by western blotting with antibodies PHF-1, MC-1, AT-8 and Tau-5 (**Fig. 6e, for blots see Extended Data Fig. 10b**). Under stress, the presence of BAG2 resulted in 3.66 ± 2.03 -fold decrease in PHF-1/Tau-5 ($p = 0.0027$), 0.69 ± 0.63 -fold decrease in MC-1/Tau5 ($p = 0.093$) and 1.13 ± 0.90 -fold decrease in AT8/Tau5 ($p = 0.041$) (Fig. 6e). Directly to the point raised by the reviewer, a key site of BAG2 action appears to be on the microtubule as after the 2h time period of hyperosmotic stress, BAG2 condensates remained associated with Tau on the microtubules (**Extended Data Fig. 10c**).

12. Fig. 7c: The authors state that bafilomycin has no effect on BAG2 condensates. However, in the presented picture BAG2 looked much more diffusely distributed and seemed to display reduced condensate formation upon bafilomycin treatment. Although BAG2-mediated degradation doesn't seem to involve autophagy, there still seems to be a crosstalk between these pathways.

Authors: *That is a good point. Indeed, we agree that a crosstalk between the BAG2-mediated degradation and autophagy very likely happens. However, regarding the image where BAG2 looked much more diffusely distributed (old main Figure 7c), the scale was misleading (insert) and that was corrected in this current version. The scale bar is now standardized for image a, b and c. Data are shown in the "main Figure 7c".*

13. Fig. 7f: The picture doesn't show p62 accumulation in regions devoid of BAG2 as stated by the authors. It rather seems that p62 is concentrated in a portion of the BAG2 condensate. Segregation of BAG2 and p62 cannot be deduced from the presented data.

Authors: *We have revised our interpretation in accord with the reviewer's comment. We changed the title of the section to "LAMP-1, p62/SQSTM1 and BAG2 condensates" instead of "Segregation of LAMP-1 and p62/SQSTM1 from BAG2 condensates" in the "result section – line 305". The description was changed in the main text "result section – lines 318-321". We also changed the "main figure 8" pointing out LAMP-1 labeling predominated in regions devoid of BAG2 signal which formed multiple small puncta throughout and p62/SQSTM1 forms a well-defined region within the locale of the BAG2 signal.*

Reviewer #2:

Carrettiero et al. investigate the formation of membraneless organelles by BAG2. Using imaging and spectroscopy they demonstrate that BAG2 containing organelles are formed under stress, that they are ubiquitin independent, that the organelles align with microtubules and that they degrade tau protein. Using Fluorescence Recovery after Photobleaching (FRAP) they demonstrate that the organelles exchange BAG2 with the cytosol. They calibrate their concentration-intensity relation by Fluorescence Correlation Spectroscopy (FCS) and then use image analysis to derive monomeric BAG2 and BAG2 condensate concentrations to determine the transfer free energy of solution and condensates. Their analysis before and after stress indicates that heterotypic interactions increase after stress in BAG2 condensates, although it is not clear what drives these changes. The manuscript does not contain sufficient information to judge the use of FCS.

Authors: *We have now included all the information necessary regarding the FCS analysis as requested by the referee:*

- ***“Source data – main Figures (sheet Fig 4a, b)”*** regarding Main Figure 4a,b

- ***“extended data – Figure 5”*** and ***“Source data - extended Figures – sheets Fig 5”***.

Further description was added in the main text “Methods section – lines 463 - 489”.

The authors should include:

- FCS calibration and measurement curves including fits, fit parameters and residuals.

Authors: *FCS calibration and measurement raw data have been included in:*

- ***“Source data – main Figures (sheet Fig 4a, b)”*** regarding Main Figure 4a,b

- ***“extended data – Figure 5”*** and ***“Source data - extended Figures – sheets Fig 5”***

*Description was added in the main text “Methods section – lines 463 - 489”. This includes raw counts, autocorrelation plots, fits of the autocorrelation plots, residuals, and fit parameters. Representative raw data and autocorrelation fits are included in as **Extended data – Figure 5**. Final fit used a Triplet Extended 3D models. For a small subset of measurements, diffusion coefficients assumed values that*

were outside of the range observed for the rest of the measurements. Forcing these to more appropriate values had no substantial effect on the $G0$ parameter, the only value that was used in subsequent analysis. These data points were excluded to avoid any possible artifacts in the data.

- the concentration intensity calibration derived from FCS

Authors: Attached below and also included in “**Extended data – Figure 5a**”. Of note, inclusion or exclusion of data at high intensities (those showing the most variance due to intrinsic errors associated with measuring higher concentrations by FCS) did not substantially change the calibration curve.

-Where FCS calibrations and measurements performed with the same fit model?

Authors: The following was added to the main text: “**Methods section – lines 463 - 489**”.

Calibrations were performed with Alexa488 dye free in solution and fit with its known diffusion coefficient to determine the effective excitation volume for the subsequent experiments. Data were fit to a triplet extended 3D model, which is appropriate for most live cell experiments.

$$G(\tau) = 1 + \sum_{i=1}^3 \frac{p_i}{1 + \tau/\tau_i} + G_{\text{eff}},$$

$$\kappa = \frac{z_0}{w_0},$$

$$V_{\text{Eff}} = \pi^{3/2} \omega_0^2 z_0,$$

$$\langle C \rangle = \frac{\langle \rangle}{V_{\text{Eff}} N_A}$$

$$D = \frac{W_0}{4}$$

$$\sum p = \frac{\langle \rangle}{-\Sigma}$$

where n is the number of independently diffusing species, p is the diffusion amplitude of the i -th diffusing species, τ is the diffusion time of the i -th diffusing species, α is the anomaly parameter of the i -th diffusing species, m is the number of triplet states (set to 1), T is the fraction of molecules in the j -th triplet state, τ is the lifetime of j -th triplet state, κ is the structural parameter for the focal volume, G is the correlation offset, N is the average number of molecules in the focal volume, V_{Eff} is the effective excitation volume (estimated by the calibration measurement), N_A is Avogadro's number, C is the concentration of molecules in the focal volume, D is the diffusion coefficient of the i -th diffusing species, ω_0 is the effective lateral focal radius at half intensity, and z_0 is the effective focal radius along the optical axis at half intensity.

-Did the cell measurements show only one diffusive component?

Authors: Yes, as described in the methods, cells expressing extremely low concentrations of BAG-GFP were used to build the calibration curve and we expected no formation of condensates that would complicate measurements. When fitting to a two-component diffusion model, the chi-squared of the fits did not significantly improve, and thus we had no reason to deviate from a single component model. There were a few measurements that fit lower than expected diffusion coefficients, but these did not correlate to higher concentrations of BAG2 or puncta like structures. Forcing these fits to appropriate diffusion coefficients had no significant effect on G_0 values that determine the concentrations used in the calibration. Regardless these points were left out of the fits to avoid any possible problems.

Minor issues:

-Line 309: Please rewrite the sentence: "... BAG2 phase separation in cells with by showing ..."

Authors: *Done. It was changed in the main text line 343.*

-Line 318 and 383: replace “Zsgreen” with “ZsGreen” for consistency.

Authors: *Done.*

Line 421: replace “ $\mu\text{M}^2/\text{sec}$ ” with “ $\mu\text{m}^2/\text{s}$ ”

Authors: *Done. We change to “ $\mu\text{m}^2 \text{s}^{-1}$ ” for consistency - lines 466*

Reviewer #3:

-The work of Carrettiero et al focuses on a BAG2 Ubiquitin-Independent Degradation condensate. The authors report that BAG2 is a marker for a membraneless organelles triggered by stress but distinct from stress granules and processing bodies. BAG2 appears to lack RNA and ubiquitin, while favors degradation of client proteins via the 20S proteasome. In particular, this new type of membraneless organelles respond specifically to hyperosmotic stress.

-Overall, the work is of significance for the field and the results appear new in the field, since previous work reported BAG2 only as a component of phase-separated organelles, but not as one of the main component associated to hyperosmotic stress granules (see Esposito et al., 2021)

Authors: *We thank the reviewer for this important reference. We added as followed: Esposito and colleagues (2021) recently reported BAG2 within a cell isolate particles, which contained about 600 other proteins. “introduction section” – lines 64, 65”.*

-The work supports the main findings that BAG2 condensation respond to hyperosmotic stress favoring phase separation, whereas arsenite-induced stress does reduce the condensation of the protein. Main markers of stress granules and processing bodies are not found in colocalization with BAG2, supporting the protein is part of a distinct type of organelle. Finally, the condensates seem to be lacking RNA content. In terms of condensation, BAG2 phase separation requires both the coiled-coil and BAG2 domain.

Authors: *We thank the reviewer for these comments.*

-There are few concerns that may need to be addressed:

-BAG2 Condensates increase in size with hyperosmotic stress. However, hyperosmotic stress causes an increase in the cell concentration by altering the volume of the cell. If that is the case, the change in size (but not in number) of the condensates would represent just a shift in the total concentration of the components. It would be helpful to provide an estimate of the variation in the cell size before and after hyperosmotic stress. If a difference in volume is observed, it can be used to estimate the phase diagram of the component as a function of the total protein concentration.

Authors: *The reviewer mentioned that hyperosmotic stress causes an increase in the cell concentration by altering the volume of the cell. Presumably the increase refers to either BAG2 or concentrations of all cell constituents unaffected by the stress. With regard to this comment, the phase behavior of protein condensation is independent of the total concentration, i.e., at two different concentrations the phase behaviors must remain unchanged (within certain limits). But what would change under volume shift is the amount of condensate such that at higher concentration we could get a greater number of the droplets and subsequently more condensates or larger condensates. The control over condensate size and condensate number is poorly understood. If we could get an estimate of volume change, we might be able to linearly scale the amount condensate before and after applying the osmotic pressure. However, this interesting direction will require extensive calibrations to capture very small condensates, 3D reconstructions, and accurate measurements of osmotic pressure over a sufficient dynamic range. Perhaps more germane to the comment is the observation that BAG2 condensates also increased in size after stresses due to temperature change (42°C), oligomycin and LPS which do not affect cellular volume **“Introduction section”- lines 94-96 and lines 103-104.** Another aspect of this question is the out of equilibrium nature of the volume transition itself, meaning that the volume transition can enhance phase separation through enhancing composition fluctuation. Thank-you for this thought-provoking comment.*

- FCS is used to determine the concentration inside and outside the dense phase of the organelles but no example of correlation curves, FCS calibration and fit is reported besides the ratio of the final concentrations. This makes impossible to judge the quality of the source data.

Authors: *FCS calibration and measurements of raw data have been included in:*

-“Source data – main Figures (sheet Fig 4a, b)” regarding Main Figure 4a,b

-“extended data – Figure 5” and “Source data - extended Figures – sheets Fig 5”

*A description was added in the main text “Methods section – lines 463 - 489”. This includes raw counts, autocorrelation plots, fits of the autocorrelation plots, residuals, and fit parameters. Representative raw data and autocorrelation fits are included in **Extended data – Figure 5**. The final fit used a Triplet Extended 3D models. For a small subset of measurements, diffusion coefficients assumed values that were outside of the range observed for the rest of the measurements. Forcing these to more appropriate values had no substantial effect on the G0 parameter, the only value that was used in subsequent analysis. Regardless, these data points were excluded to avoid any possible artifacts in the data.*

-It would be also important to report the observed changes in the diffusion coefficient and number brightness since they report about the local viscosity perceived in the cell, possible aggregation or quenching effects.

Authors: *Diffusion when measured in dilute, low-expression conditions did not change. It is beyond FCS capabilities to measure the diffusion coefficient in condensates. There are multiple technical reasons for this with the most prohibitive being that the concentrations inside condensates are far too high to make FCS measurements. We made no attempts to discuss diffusion and viscosity in our system for this reason. However, due to advances in the linearity of detector responses, it is possible to create an intensity/concentration calibration curve, and make claims about thermodynamic properties. This is what we have done in this work.*

-Finally, the reference to the method is actually making use of specific TAG-lens that the authors do not mention and that possibly is not applied to their experiments. Adding or substituting the citation for other works that harness FCS to study biomolecular condensates may provide a more reliable sources.

Authors: *We have substituted and added more references as suggested by the referee.*

Ref 23, 24 (line 183)

Ref 23, 24, 58 (line 467)

The following Reference was excluded:

Politi, A.Z. et al. Quantitative mapping of fluorescently tagged cellular proteins using FCS-calibrated four dimensional imaging. Nat Protoc 13, 1445-1464 (2018).

- When plotting the dependence of the DeltaG on the dilute concentration, one would expect the values to tend to zero at low and high protein concentration, since the limiting conditions coincide with the transition to a single phase. One would expect this to be true even if the phase separation is heterotypic, unless BAG2 is recruited to the condensate. The authors may want to expand on their reasoning on how that type of curvature can be obtained and what it implies in terms of the condensate composition and role of BAG 2. E.g. is BAG2 a client or a scaffold in these condensates?

Authors: *This is a very interesting but challenging question. It would be nice to have a truly single “dense” phase, but we have never observed this extreme in the cell. We are solidly in the two-phase regime it seems. We believe that BAG2 is indeed a scaffold but until we know exactly all the components in the BAG2 condensate it would be hard to precisely draw that conclusion.*

- Have the authors considered changing temperature to explore the phase diagram of the protein? It seems that FCS provides boundaries for both the dense and light phase. Temperature, which sets the strength of interactions, should provide a mean to see how these concentrations changes and to identify whether increasing or decreasing temperature is favoring the phase separation.

Authors: *This is a very interesting topic currently under investigation in separate work in our lab. In general, one would expect that by increasing the temperature the phase separation tendency would increase; however, it is very challenging to determine exactly what temperature does to phase separation mechanism, because no info is available regarding the temperature response of the protein structure. Furthermore, heat is a stressor that triggers other biological events that could inhibit or enhance phase separation independently of enthalpic consideration. This is seen when heat stress is applied. We observed increase in BAG2 phase separation at 42°C in “**Extended Figure 1d and Extended Figure 2a**”*

- FRET constructs for BAG2 and tau interaction detail a peculiar territorial organization of the components in the dense phase. Is this stable over time or does it change?

Authors: Thank you for this interesting question which is certainly a future direction using super resolution microscopy. At this juncture, a conservative interpretation of the data makes it hard to infer a territorial organization of the components in the BAG2 dense phase at different time points after stress. The territory occupied by BAG2 condensates (“**main Figure 6d – image**”) is very close to or below the maximum resolution of the microscope used ($0.15\mu\text{m}$). And the FRET measurements were performed in the whole BAG2 condensates ($\sim 0.8\mu\text{m}$). To measure and compare the macrodomain of FRET signal inside the BAG2 condensates at different time points after stress we would require much higher resolution. However, indirect results suggest that the territorial organization of the components in the dense phase after stress appear to be stable for a least 2h and remain aligned to MT for at least 2h “**Extended Figure 10c**”. After 2h of osmotic stress, the accumulation of phospho-Tau species decreased in presence of BAG2 “**main figure 6e**”, suggesting that the degradation promoted by the co-chaperone BAG2 was still operative.

- Are the BAG2 condensates specific to the studied cell line? or do they appear in other cell lines too?

Authors: They also appear in other cell lines. We have included endogenous BAG2 staining after sucrose treatment using H4 cell in “**Main Figure 3i**”. We also observed them in HeLa cells (see below).

- Is the response of BAG2 condensate to hyperosmotic stress specific to sucrose or other osmolytes can cause the same effect? For example, addition of small PEG molecules (e.g PEG 400) has been shown to cause collapse of cells - does this provide a similar result than sucrose?

Authors: *Yes, we observed increased endogenous BAG2 condensates in HeLa after 30min of PEG 8000 30% and Sorbitol 300mM as shown below. We did not test small PEG molecules such as PEG 400.*

- When mentioning that oligomerization would lower the barrier for LLPS (line 126) the authors may want to explicit their reasoning (e.g. increased multivalency).

Authors: *We have made the reasoning explicit in the text “**result section – lines 128 - 132**”. In essence, the LLPS is more likely initiated through a heterogeneous nucleation on the surface of the oligomer complexes. As it is known from classical nucleation theory, the free-energetic barrier of heterogeneous LLPS is lower than homogeneous LLPS where the oligomers are absent, and subsequently the rate of heterogeneous nucleation will be faster. In terms of the set of interactions that exist in the driving force of LLPS, the oligomers in solution enhance LLPS due to the fact that the multivalent and heterotypic interactions are strongly increased in presence of the oligomers.*

REVIEWER COMMENTS

Reviewer #1 (Remarks to the Author):

The revised version of the manuscript addresses my previously expressed concerns.

Reviewer #2 (Remarks to the Author):

The authors have provided now the FCS data and figures to which I have some comments. In the present form the absolute concentration values are not reliable and thus might lead to wrong values for the transfer free energies. The relative differences might still persist even if the absolute values change. But a new measurement of the calibration should be conducted and it should be tested, if some of the issues are instrumental issues, whether this influenced the rest of the data.

1. The correlation function(extended Fig 5b panel 2) has a dip around 70 ms. The dip is also prominent in the residuals (panel 3). Is that dip present in all measurements, in particular in the calibration with Alexa488? This points toward a correction of the data or some artifact which leads to bad fits as can be seen in the figure.

2. Was the data background corrected? If yes, the fit should (be forced) to pass the origin. If not, there might be some deviation from real concentrations.

3. What are the intensity units on the graph in extended Fig. 5a?

4. In the Excel file for that figure, the χ^2 values differ strongly. I assume these are reduced χ^2 values? This should be clarified. And have the authors tried to fit with more complex models or how can the large χ^2 values be explained?

5. The structure parameter of the experimetns is 1.3, this is lower than the theoretical limit. A factor of 3-8 is typically expected. How did the authors verify that value? Also, the lateral radius seems very large, much larger than the diffraction limit (what was the pinhole size?) and the axial radius very small. Authors should verify the data. Either there is a problem in data fitting, or the system might be misaligned.

6. The diffusion time for Alexa 488 is very high with 140 us. And for the other measurements the diffusion times vary strongly, from 2,700 to 24,000 us. This might indicate that not only a single species, but multiple species are measured? The authors could evaluate the calibration for data with different diffusion times separately. E.g. most ACFs seem to be between 3-5,000 us. Do those give the same calibration?

7. The authors indicate that they used a fit with an anomaly factor (page 23), but that data is not given in Excel.

Minor issues:

Page 23: The lateral and axial radii are defined over the drop of the intensity to e^{-2} of its maximum value no the half-intensity.

Reviewer #3 (Remarks to the Author):

The authors have largely addressed my concerns.

I thank you them for taking in consideration my suggestions and discussing them.

One point remains very important to me and this regards the FCS data and their presentation. I think the approach is valuable and I hope my suggestion can help strengthening a key-point in the interpretation of the results.

To judge the extrapolated value for FCS, it is important to show a comparison between the correlation curves for the light and dense phase and to compare the concentration vs intensity obtained in the dense phase with the calibration measurements. Similarly, the authors mention deviations in the diffusion coefficient of some of the measured points. A plot of the diffusion coefficients should provide information of the diffusion of the protein in the dense phase. Finally, extended Figure 5a does not reports units for intensity, which perhaps are counts.

NCOMMS-21-24463A

In response to the reviewers' comments, we have made revisions to our manuscript. We thank for the in-depth review of our manuscript and the constructive criticism helping us to improve the quality of our work.

Below it is a point-by-point responses raised:

Reviewers' comments:

Reviewer #1:

The revised version of the manuscript addresses my previously expressed concerns.

Reviewer #2:

The authors have provided now the FCS data and figures to which I have some comments. In the present form the absolute concentration values are not reliable and thus might lead to wrong values for the transfer free energies. The relative differences might still persist even if the absolute values change. But a new measurement of the calibration should be conducted and it should be tested, if some of the issues are instrumental issues, whether this influenced the rest of the data.

*We greatly appreciate the reviewers in depth comments. They have prompted us to communicate with instrument specialists at Leica to ensure that our measurements and analysis are of the quality required to calculate the transfer free energies in the rest of our analysis. During our initial data collection, we made multiple measurements for the calibration and upon closer inspection, and at the direction of the reviewer, realized some errors in our fitting procedures that produced artifacts. Below we have responded to each point the reviewer suggested and feel confident that our analysis is now robust. The revision does not change any of the conclusions from the FCS data. We have updated our figures to reflect these changes (**Main fig. 4a,b; Extended fig 5a,b and respective Source data file**).*

It should be noted—as the reviewer has pointed out—that the relative differences between our measurements would persist even if the absolute values changed. The large dynamic range of our detectors allow for the use of our linear calibration curve (derived from these FCS measurements) to be extrapolated to the larger intensities found in the Bag2 droplets. Further, because the transfer free

energies are unitless, they are completely independent of this calibration (at least in the case where the y-intercept of the calibration approaches zero). What FCS gives us is the ability to put estimates on the concentrations of the dilute phase, something that could be reported as relative intensities. We are pleased that our FCS analysis is vastly improved prompted by the reviewer's comments but would add that the FCS concentration versus intensity curve is not necessary for any of the transfer free energy analyses. What the FCS concentration versus intensity curve does, is give us a handle on the concentration ranges over which these processes are occurring, which is a unique and informative set of data, but it does not affect the arguments about heterotypic interactions occurring during Bag2 phase separation.

1. The correlation function (extended Fig 5b panel 2) has a dip around 70 ms. The dip is also prominent in the residuals (panel 3). Is that dip present in all measurements, in particular in the calibration with Alexa488? This points toward a correction of the data or some artifact which leads to bad fits as can be seen in the figure.

After examining the entire dataset, we found that the dip was only present in a subset of the measurements and was not present in any of the Alexa488 measurements. Looking closer at the subset of data that had a dip, we were able to determine that it was the result of our software's bleach correction overcompensating for signal attenuation. To address this, we completely removed bleach correction because for most of our data, there was not strong bleaching of the fluorophore. For data that did have bleaching, most of it occurred within the first seconds of the measurement. As such, removing this part of the curve vastly improved our fits. We have provided new fit results and have updated extended Fig 5b to show this change.

2. Was the data background corrected? If yes, the fit should (be forced) to pass the origin. If not, there might be some deviation from real concentrations.

The fitting of our FCS data showed a near zero background contribution. For the calibration curve, we have measured the background from our images and subtracted this from the data (the background values were all near zero). The fit is now forced through zero.

3. What are the intensity units on the graph in extended Fig. 5a?

To construct the concentration calibration curve, we first recorded an image of the cell using a set of fixed parameters. The intensity units are the number of photons that were recorded at the pixel that corresponded to the pixel that the FCS measurement was made at. We changed to "Counts" to better represent our FCS measurement.

4. In the Excel file for that figure, the χ^2 values differ strongly. I assume these are reduced χ^2 values? This should be clarified. And have the authors tried to fit with more complex models or how can the large χ^2 values be explained?

*These are an AIC χ^2 value used for the selection of an appropriate model. We have tried to fit more complex models (such as multiple diffusing species, and have seen minor improvements in the χ^2 , but not large enough to justify moving to more complex models. We have decided to replace the χ^2 we have reported with a reduced χ^2 to avoid confusion (**Source Data - Extended Fig 5a,b – Fitparameters**). The few datasets with high reduced χ^2 have not been included in the calibration curve.*

5. The structure parameter of the experiments is 1.3, this is lower than the theoretical limit. A factor of 3-8 is typically expected. How did the authors verify that value? Also, the lateral radius seems very large, much larger than the diffraction limit (what was the pinhole size?) and the axial radius very small. Authors should verify the data. Either there is a problem in data fitting, or the system might be misaligned.

We have multiple measurements of our Alexa488 calibration and after looking at other measurements we have concluded that the fit we used had too much noise to be reliable. As the system

had been calibrated and aligned at the same calendar week the experiments were run and the objective collar had been optimized the day of measurements, we are confident that the system was well tuned. We now report a structural parameter of 2.99, a lateral radius of 0.247, and an axial radius of 0.740. Discussions with instrument scientists at Leica have confirmed that these values are not unexpected for our microscope configuration.

6. The diffusion time for Alexa 488 is very high with 140 us. And for the other measurements the diffusion times vary strongly, from 2,700 to 24,000 us. This might indicate that not only a single species, but multiple species are measured? The authors could evaluate the calibration for data with different diffusion times separately. E.g. most ACFs seem to be between 3-5,000 us. Do those give the same calibration?

The Alexa488 diffusion time in our fitting is now 33 μ s. The average measurement for the protein is now $3300 \pm 1100 \mu$ s. We believe that the previous range was a result of poor fitting. It is possible that in some cases larger species may pass through the excitation volume, but we attempted to make measurements of cells with low concentrations of Bag2 construct as to try and avoid its oligomerization into larger species.

7. The authors indicate that they used a fit with an anomaly factor (page 23), but that data is not given in Excel.

Thank you this was an error on our part, the parameter has been allowed to float and this has also improved the quality of the fits (Source Data - Extended Fig 5a,b – Fitparameters).

Minor issues:

Page 23: The lateral and axial radii are defined over the drop of the intensity to e^{-2} of its maximum value no the half-intensity.

Fixed (main text, line 482)

Reviewer #3.

The authors have largely addressed my concerns.

I thank you them for taking in consideration my suggestions and discussing them.

One point remains very important to me and this regards the FCS data and their presentation. I think the approach is valuable and I hope my suggestion can help strengthening a key-point in the interpretation of the results.

To judge the extrapolated value for FCS, it is important to show a comparison between the correlation curves for the light and dense phase and to compare the concentration vs intensity obtained in the dense phase with the calibration measurements.

We thank the reviewer for this comment. However, it is impossible to measure the diffusion coefficient in the dense phase. FCS is very limited in the concentration range to which it is accessible. As the concentration of the molecule of interest goes up, the $G(0)$ —the y-intercept— goes to zero. Practically this means that the condensates have concentrations at least an order of magnitude higher than can be measured with FCS.

Similarly, the authors mention deviations in the diffusion coefficient of some of the measured points. A plot of the diffusion coefficients should provide information of the diffusion of the protein in the dense phase.

We have addressed deviations for the diffusion coefficients by improving our fittings. While we agree that it would be extremely interesting to measure the diffusion of Bag2 in the dense phase, it is not feasible.

Finally, extended Figure 5a does not reports units for intensity, which perhaps are counts.

Yes, fixed.

REVIEWERS' COMMENTS

Reviewer #2 (Remarks to the Author):

The authors have corrected the FCS data and have answered all my questions.

Reviewer #4 (Remarks to the Author):

The authors have addressed all the main concerns. However, the part concerning FCS and its use to compute the change in the partitioning free energy still contain some possible source of concern. While this does not substantially impact the scope of the paper, I think it is important to provide a precise indication of the methodology to facilitate the use of the same approach by other users as well as cautionary notes when the interpretation may be subject to experimental errors that cannot be precisely estimated.

The additional data and explanation in the response to reviewers clarifies that FCS is used only to measure concentration of molecules in the dilute phase. While this establish a somehow linear relationship between counts and number of molecules, there is no direct evidence that a similar relation may persist in the dense phase (even if the detectors have a linear response at that specific detection). This is very difficult to disentangle and test within a living cell. The authors may want at least to mention the possible limitations of their approach in performing these estimates.

There is no discussion on the fact that the diffusion coefficient and anomaly diffusion term vary substantially across the calibration curve. Some of these data would benefit from being plotted in one of the extended figures instead of being provided as a supplemental string of data that the reader has to plot to evaluate. Though the variability may simply arise from transient interactions, it is still important to provide information about the fact that such transient interactions are present. The lack of a correlation between the concentration of molecules and the diffusion coefficient help to support that fluctuations are due to heterotypic interactions and not homotypic interactions.

The authors may want to clearly present the FCS formula that has been used to fit their data. If the one used to analyze the data is the one with a single diffusing species, the authors may want just to provide that equation and neglect terms for additional species.

Finally, it is unclear how the new values for the aspect ratio of the detection volume have been determined. They are presented with three significant digits (a structural parameter of 2.99, a lateral radius of 0.247 μm , and an axial radius of 0.740 μm). However, it seems very difficult to achieve a precision on the third significant digit, perhaps even on the second one. Finally, the values should be reported in the main text or extended data. Currently, unless I have missed them in the text, they are reported only in the response to the reviewer and in the analysis file.

NCOMMS-21-24463C

In response to the reviewers' comments, we have made revisions to our manuscript. We thank for the in-depth review of our manuscript and the constructive criticism helping us to improve the quality of our work.

Below it is a point-by-point responses raised:

Reviewers' comments:

Reviewer #4:

Reviewer: The authors have addressed all the main concerns. However, the part concerning FCS and its use to compute the change in the partitioning free energy still contain some possible source of concern. While this does not substantially impact the scope of the paper, I think it is important to provide a precise indication of the methodology to facilitate the use of the same approach by other users as well as cautionary notes when the interpretation may be subject to experimental errors that cannot be precisely estimated.

The additional data and explanation in the response to reviewers clarifies that FCS is used only to measure concentration of molecules in the dilute phase. While this establish a somehow linear relationship between counts and number of molecules, there is no direct evidence that a similar relation may persist in the dense phase (even if the detectors have a linear response at that specific detection). This is very difficult to disentangle and test within a living cell. The authors may want at least to mention the possible limitations of their approach in performing these estimates.

Answer: Thanks for pointing that out. The following sentence was added to the text in order to address the possible limitations.

"It is important to note that several factors can have large effects on the concentration estimates, leading to an accuracy with 2-fold of the actual values (Bracha et al., 2018). These effects include photobleaching, protein maturation times, and other optical artifacts including cover-slip thickness variation, optical saturation, and other aberrations (Loman et al, 2018; Petrásek and Schwille, 2008)."
(See page 23, line 486)

Reviewer: There is no discussion on the fact that the diffusion coefficient and anomaly diffusion term vary substantially across the calibration curve. Some of these data would benefit from being plotted in one of the extended figures instead of being provided as a supplemental string of data that the reader has to plot to evaluate. Though the variability may simply arise from transient interactions, it is still important to provide information about the fact that such transient interactions are present. The lack of a correlation between the concentration of molecules and the diffusion coefficient help to support that fluctuations are due to heterotypic interactions and not homotypic interactions.

Answer: The following plot was added to the extended figure 5 as a new panel c and a discussion of this was inserted at the main text (See page 10, line 192).

Page 10, line 192: *“Further, when fitting individual FCS traces fluctuations in the measured diffusion coefficients were uncorrelated with measured concentrations, supporting the hypothesis that heterotypic interactions are playing an important role in this system (Extended Data Fig. 5c).”*

We have also included the example plot below in the extended figure 5b:

When plotting other traces, no visible difference in the fits is apparent, we believe plotting the concentration versus the diffusion coefficient as the reviewer has suggested nicely shows that there is no correlation between these variables.

Reviewer: The authors may want to clearly present the FCS formula that has been used to fit their data. If the one used to analyze the data is the one with a single diffusing species, the authors may want just to provide that equation and neglect terms for additional species.

Answer: Fixed to include just the single diffusing species FCS formula (Please, see page 23, line 469).

$$G(\tau) = G(0) \left[1 - T + T e^{-\frac{\tau}{\tau_T}} \right] \left[1 + \left(\frac{\tau}{\tau_D} \right)^\alpha \right]^{-1} \left[1 + \frac{1}{\kappa^2} \left(\frac{\tau}{\tau_D} \right)^\alpha \right]^{-\frac{1}{2}} + G_\infty$$

$$\kappa = \frac{z_0}{\omega_0},$$

$$V_{Eff} = \pi^{3/2} \omega_0^2 z_0,$$

$$\langle C \rangle = \frac{\langle N \rangle}{V_{Eff} N_A},$$

$$D = \frac{\omega_0^2}{4\tau_i},$$

where τ_D is the diffusion time, α is the anomaly parameter, τ_T is the lifetime of the triplet state, T is the triplet amplitude, κ is the structural parameter for the focal volume, G_∞ is the correlation offset, N is the average number of molecules in the focal volume, V_{Eff} is the effective excitation volume (estimated by the calibration measurement), N_A is Avogadro's number, C is the concentration of molecules in the focal volume, D is the diffusion coefficient,

ω_0 is the effective lateral focal radius at e^{-2} of its maximum value, and z_0 is the effective focal radius along the optical axis at e^{-2} of its maximum value.

Reviewer: Finally, it is unclear how the new values for the aspect ratio of the detection volume have been determined. They are presented with three significant digits (a structural parameter of 2.99, a lateral radius of 0.247 μm , and an axial radius of 0.740 μm). However, it seems very difficult to achieve a precision on the third significant digit, perhaps even on the second one. Finally, the values should be reported in the main text or extended data. Currently, unless I have missed them in the text, they are reported only in the response to the reviewer and in the analysis file.

Answer: in order to address this issue, we have added the following sentence to the text (See page 23, line 465).

“After fixing the diffusion coefficient for Alexa 488 calibrations, a structural parameter of 3.0, a lateral radius of 0.25 μm , and an axial radius of 0.74 μm were fit.”